# On the Limits of Sparse Autoencoders: A Theoretical Framework and Reweighted Remedy

**Jingyi Cui**[1][*]     **Qi Zhang**[1][*]     **Yifei Wang**[2]     **Yisen Wang**[1,3][†]
[1]State Key Lab of General AI, School of Intelligence Science and Technology, Peking University
[2]Amazon AGI SF Lab[‡]     [3]Institute for Artificial Intelligence, Peking University

## Abstract

Sparse autoencoders (SAEs) have recently emerged as a powerful tool for interpreting the features learned by large language models (LLMs). By reconstructing features with sparsely activated networks, SAEs aim to recover complex superposed polysemantic features into interpretable monosemantic ones. Despite their wide applications, it remains unclear under what conditions SAEs can fully recover the ground truth monosemantic features from the superposed polysemantic ones. In this paper, we provide the first theoretical analysis with a closed-form solution for SAEs, revealing that they generally fail to fully recover the ground truth monosemantic features unless the ground truth features are extremely sparse. To improve the feature recovery of SAEs in general cases, we propose a reweighting strategy targeting at enhancing the reconstruction of the ground truth monosemantic features instead of the observed polysemantic ones. We further establish a theoretical weight selection principle for our proposed weighted SAE (WSAE). Experiments across multiple settings validate our theoretical findings and demonstrate that our WSAE significantly improves feature monosemanticity and interpretability.

## 1 Introduction

The "black-box" nature is a long-standing problem plaguing the mechanistic interpretability of deep neural networks. One of the key problems is feature *polysemanticity*, where an individual neuron is often activated by multiple semantically unrelated features (Elhage et al., 2022b; Hänni et al.; Scherlis et al., 2022; Zhang et al., 2023; 2025). This issue is particularly evident in large language models (LLMs) as neurons rarely correspond to distinct and well-defined features. Previous studies have proposed the *superposition* hypothesis that the polysemantic features are linear combinations of the *monosemantic* ones, so that models can represent more features than they have dimensions (Elhage et al., 2022b; Hänni et al.; Scherlis et al., 2022).

To disentangle the superposed polysemantic features, sparse autoencoders (SAEs) (Gao et al., 2025; Makhzani & Frey, 2013; Minegishi et al., 2025; Ng et al., 2011) have demonstrated significant potential for identifying interpretable monosemantic features. They have been widely used as a promising approach to enhancing the interpretability of LLMs (Cunningham et al., 2023; Ferrando et al., 2025; Gao et al., 2025; Lieberum et al., 2024) and VLMs (Daujotas, 2024; Lim et al., 2025; Lou et al., 2025; Thasarathan et al., 2025). Typically, an SAE has a single wide layered encoder-decoder architecture, with sparse activations such as ReLU (Cunningham et al., 2023), JumpReLU (Rajamanoharan et al., 2024b), TopK (Gao et al., 2025), BatchTopK (Bussmann et al., 2024), etc. While previous works mainly focused on studying architecture (Braun et al., 2024; Makhzani & Frey, 2013; Rajamanoharan et al., 2024a; Tibshirani, 1996; Bussmann et al., 2025) or evaluation (Minegishi et al., 2025; Karvonen et al., 2025) of SAEs, the theoretical understanding of the identifiability of SAEs is still lacking. Specifically, we wonder: *Can SAEs recover the ground truth monosemantic features from the polysemantic inputs?*

To answer the above question, in this paper, we investigate the theoretical conditions of SAE feature recovery. First of all, under the superposition hypothesis, we propose a theoretical framework and

---

[*]Equal contribution
[†]Corresponding author: Yisen Wang (yisen.wang@pku.edu.cn)
[‡]This work was completed at MIT prior to Yifei Wang joining Amazon.

derive the closed-form optimal solution to SAEs. Nonetheless, we show that a full recovery of the ground truth monosemantic features is not theoretically guaranteed under general conditions, with standard SAEs plagued by feature shrinking and feature vanishing. As a possible explanation why SAEs do work well in some empirical cases, we further show that the sparsity of the ground truth features might be the key. Specifically, if the ground truth monosemantic features are extremely sparse, the optimal solution to an SAE is unique and precisely recovers the ground truth features. Moreover, considering that the sparsity of the ground truth features is not controllable through training, when the extreme sparsity assumption is not met (SAE feature not fully recoverable), we propose a reweighting strategy to enhance the reconstruction of the ground truth features. We also propose a principle for the weight selection through theoretical analysis. Specifically, we derive the theoretical relationship between the loss of SAE feature reconstruction and that of the ground truth feature reconstruction, and discuss that smaller weights on the more polysemantic dimensions help reduce the negative interferences hindering the reconstruction of ground truth features.

Our contributions are summarized as follows.

- We propose a theoretical framework for analyzing SAE feature recovery based on the superposition hypothesis and derive a closed-form solution to SAEs. Based on this, we show that SAEs fails to recover the ground truth monosemantic features in general unless the ground truth features are extremely sparse.

- In low sparsity conditions where SAE full recovery does not meet, we propose a reweighting strategy to improve the reconstruction of ground truth features by SAEs and theoretically discuss the principle of weight selection.

- We validate our theoretical findings through experiments and show that our reweighting strategy significantly enhances the monosemanticity and interpretability of SAE features.

## 2 RELATED WORKS

### 2.1 POLYSEMANTICITY AND SUPERPOSITION HYPOTHESIS.

A widely accepted explanation for feature polysemanticity is the superposition hypothesis (Elhage et al., 2022b; Hänni et al.; Scherlis et al., 2022), which regards a polysemantic dimension to be an approximately linear combination of several natural semantic concepts. To investigate the mechanism of polysemanticity, mainstream studies reveal that superposition occurs when a model represent more features than they have dimensions. Specifically, Elhage et al. (2022b) introduced a single-layer toy model demonstrating that polysemanticity occurs when ReLU networks are trained on synthetic data with sparse input features. Hänni et al. proved that superposition is actively helpful for efficiently accomplishing the task of emulating circuits via mathematical models of computation. Scherlis et al. (2022) explained polysemanticity through the lens of feature capacity, suggesting that the optimal capacity allocation tends to polysemantically represent less important features. Aside from the dimension restriction, Lecomte et al. (2024) demonstrated that polysemanticity can also incidentally occur because of sparse regularization or neural noise.

In this paper, we adopt the superposition hypothesis to simulate the generation of polysemantic features from the ground truth monosemantic inputs.

### 2.2 SPARSE AUTOENCODERS (SAEs).

Sparse autoencoder (SAEs) automatically learn features from unlabeled data, typically trained with sparsity priors such as ReLU activation and $l$-1 penalty (Ng et al., 2011). To mitigate the feature suppression caused by $l$-1 regularization (Tibshirani, 1996; Wright & Sharkey, 2024), Makhzani & Frey (2013) proposed k-sparse autoencoders, which replace the $l$-1 penalty with a Top-$k$ activation function. To address the same problem, Rajamanoharan et al. (2024a) proposed the gated SAE that decouples detection of which features are present from estimating their magnitudes. Besides, to ensure the features learned are functionally important, Braun et al. (2024) propose to optimize the downstream KL divergence instead of reconstruction MSE. Moreover, by using $k$-sparse autoencoders, Gao et al. (2025) found clean scaling laws with respect to autoencoder size and sparsity. Minegishi et al. (2025) proposed a suite of evaluations for SAEs to analyze the quality of monosemantic features

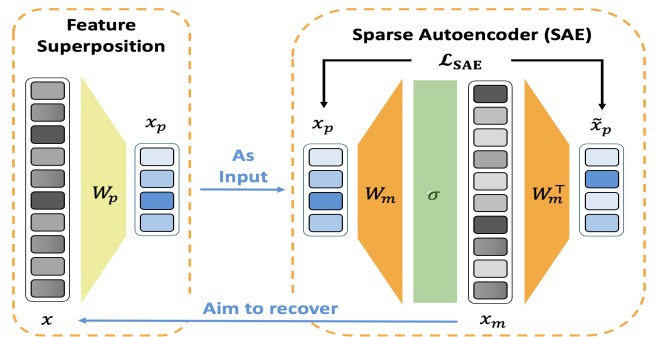

Figure 1: Theoretical framework for sparse autoencoder (SAE) feature recovery. The superposed polysemantic features $x_p$, composed of ground truth monosemantic features $x$ with matrix $W_p$, serve as the input to the SAE. For the SAE, $W_m$ denotes the weight matrix, $\sigma$ denotes the sparse activation function, and $\mathcal{L}_{\text{SAE}}$ denotes the reconstruction loss of $x_p$. Ideally, we expect the SAE output $x_m$ to fully recover the ground truth monosemantic features $x$ through reconstruction of $x_p$.

by focusing on polysemous words. They showed that compared with alternative activations such as JumpReLU and Top-$k$, ReLU still leads to competitive semantic quality.

In this paper, we investigate the identifiability of SAEs. Specifically, we are interested in the conditions under which SAEs can uniquely recover the ground truth monosemantic features from the polysemantic inputs. We consider the reconstruction-based SAEs with ReLU or Top-$k$ activations.

## 3 PRELIMINARIES & MATHEMATICAL FORMULATIONS

**Notations.** Denote $x := (x_1, \ldots, x_n)^\top \in \mathbb{R}^n$ as the ground truth monosemantic feature with dimension $n > 0$, and $x_p \in \mathbb{R}^{n_p}$ as the superposed polysemantic feature, where we assume $n > n_p > 0$. Moreover, we use $x_m \in \mathbb{R}^{n_m}$ to denote the feature learned by sparse autoencoder, where $n_m > n_p$. For the weight matrices, we denote $W_p \in \mathbb{R}^{n_p \times n}$ and $W_m \in \mathbb{R}^{n_m \times n_p}$. Throughout this paper, $\|\cdot\|$ denotes the $l_2$ norm if not otherwise specified. For mathematical conciseness, we denote $\mathbf{1}$ as the all-one vector, and $\mathbf{0}$ as the all-zero vector. Also, we denote $[n] = \{1, \ldots, n\}$. Furthermore, for a matrix $W$ we use $W_{[i,:]}$ to denote the $i$-th row of $W$, and use $W_{[:,j]}$ to denote the $j$-th column of $W$. For a vector $x$, we use $x_i$ to denote the $i$-th element of $x$.

**Sparsity of ground truth features.** For the ground truth monosemantic features $x := (x_1, \ldots, x_n)^\top$, we assume the $x_i$'s are identically and independently distributed, where given the sparse factor $S \in [0, 1]$, we have $x_i > 0$ with probability $(1 - S)$ and $x_i = 0$ with probability $S$, for $i \in [n]$. The sparsity assumption follows from Elhage et al. (2022b) and has been observed through empirical evidence (Olah et al., 2020; Elhage et al., 2022a), but ours is weaker since we do not require the uniform distribution assumption of Elhage et al. (2022b) when $x_i > 0$.

**Superposition of features.** Superposition has been widely hypothesized in previous works (Arora et al., 2018; Olah et al., 2020; Elhage et al., 2022b). Following the superposition hypothesis, we formulate the superposed polysemantic feature as a linear transformation of the ground truth monosemantic input feature $x$. Specifically, we assume the polysemantic features to be

$$x_p = W_p x \tag{1}$$

given the ground truth monosemantic features $x$ with some weight matrix $W_p$. Moreover, as discussed in Elhage et al. (2022b), the superposed dimensions usually have negative interferences, i.e., $W_{p,[:,i]}^\top W_{p,[:,j]} \leq 0$ for $i \neq j$, and form geometric sturctures of digons or polygons.

**Sparse autoencoders (SAEs).** We use a sparse autoencoder to recover the ground truth monosemantic feature $x$ from the polysemantic input $x_p$. SAE is a neural network with a single hidden layer and a sparse activation function $\sigma : \mathbb{R}^{n_m} \to \mathbb{R}^{n_m}$ such as ReLU (Cunningham et al., 2023), JumpReLU (Rajamanoharan et al., 2024b), Top-$k$ (Gao et al., 2025), etc. For mathematical simplicity, we omit the bias term and define the encoder and decoder as

$$x_m = \sigma(W_m x_p) \tag{2}$$

$$\tilde{x}_p = W_m^\top x_m. \tag{3}$$

It is trained with a reconstruction loss

$$\mathcal{L}_{\text{SAE}}(W_m; x_p) = \mathbb{E}_{x_p} \|x_p - \tilde{x}_p\|^2 = \mathbb{E}_x \|W_p x - W_m^\top \sigma(W_m W_p x)\|^2. \tag{4}$$

As not all SAE architectures include the sparsity regularization, e.g. Gao et al. (2025), we here omit the $l_1$ penalty and focus on the reconstruction loss only. Then under the optimal weight matrix

$$W_m^* = \arg\min_{W_m} \mathcal{L}_{\text{SAE}}(W_m; x_p) \tag{5}$$

the learned monosemantic feature is $x_m = \sigma(W_m^* x_p)$.

**Feature Recovery.** Ideally, we expect the SAE-learned feature $x_m$ to exactly recover the ground truth $x$, i.e., $x_m = x$ if $n_m = n$. Nonetheless, as the ground truth dimension is unknown, we believe additional zero-valued dimensions and dimension reordering are also acceptable. Therefore, in this paper, we say an SAE fully recovers the ground truth monosemantic features $x$ by reconstructing polysemantic features $x_p$ if

$$x_m \sim x, \qquad \text{if } n_m = n \text{ and} \tag{6}$$

$$x_m \sim (x, \mathbf{0})^\top, \qquad \text{if } n_m > n, \tag{7}$$

where we denote $x' \sim x$ if $x' = x$ after a reordering of the feature indexes, i.e., there exists a matrix $I^*$ formed by swapping rows of the identity matrix, such that $x' = I^* x$. Or equivalently, from the perspective of the loss function, we say an SAE fully recovers the monosemantic features if the solution $W_m^* = \arg\min_{W_m} \mathcal{L}_{\text{SAE}}(W_m; x_p)$ recovers the ground truth monosemantic features up to index reordering and zero padding.

The overall theoretical framework for SAE feature recovery is illustrated in Figure 1.

## 4 THEORETICAL RESULTS

In this section, we discuss the provable conditions for SAE feature recovery. First of all, in Section 4.2, by deriving a closed-form solution to SAEs, we show that in general cases, an SAE may fail to fully recover the ground truth monosemantic features, plagued by problems such as feature shrinking and feature vanishing. Then in Section 4.1, we show that one possible reason why an SAE still works in some real cases is the sparsity of the ground truth monosemantic features. We show that under the extreme sparsity condition, the unique optimal solution to an SAE fully recovers the ground truth features. In order to overcome such limitations of SAEs, in Section 4.3, we propose a reweighted version of SAE to achieve a better reconstruction of the ground truth monosemantic features, and theoretically discuss the weight selection principle. All proofs can be found in Appendix A. In addition, we present more discussions and extensions of our theoretical framework in Appendix B.

### 4.1 SAEs FAIL TO RECOVER GROUND TRUTH FEATURES IN GENERAL

By the mathematical formulations introduced in Section 3, we derive the closed-form solution to SAEs in Theorem 1. Note that the assumption of $W_p$ follows from the feature geometry of superposition observed in Elhage et al. (2022b), where the columns of $W_p$ within the superposed dimensions form digons or polygons.

**Theorem 1** (Closed-Form Solution to SAEs). *Let $\mathcal{L}_{\text{SAE}}$ be defined in equation 4 with sparse activation function $\sigma$. If $n_m \geq n$ and the columns of $W_p$ within the superposed dimensions form digons/polygons, then we have $W_m^* = I^*(W_p, \mathbf{0})^\top \in \arg\min_{W_m} \mathcal{L}_{\text{SAE}}(W_m; x_p)$, where $I^*$ is formed by swapping row $i$ with row $j \neq i$ of the identity matrix.*

Theorem 1 shows that given the superposition matrix $W_p$, its transpose $W_p^\top$ serves as the closed-form solution to SAEs in the sense of zero-padding and row-reordering. Specifically, the SAE recovered features are $x_m = \sigma(W_p^\top x_p)$ with zero-padding and index-reordering.

However, by such a solution, we discuss that the recovered features $x_m$ do not always recover the ground truth monosemantic feature $x$. Instead, they often deviate from the ground truth because of the *feature shrinking* and *feature vanishing* phenomena, which we illustrate in the following examples.

**Feature shrinking.** The SAE-recovered features from the polysemantic dimensions are shrunk to be smaller compared to their ground truth value. Typically, we say a superposed feature dimension is "more polysemantic" if it has interference with more ground truth monosemantic features. Then the more polysemantic a feature dimension is, the more severe its SAE-recovered monosemantic features are shrunk in value. In Example 1, as the second and third dimensions of $x$ are more polysemantic in the superposed feature $x_p$, they shrink severely after recovered by an SAE (from $(1.0, 0.8)$ to $(0.2, 0)$), whereas the first dimension, which is monosemantic, does not shrink. This could lead to incorrect interpretation of the feature, because the top-1 activated dimension of the ground truth $x$ should be the second dimension $(1.0)$, whereas when using an SAE, the top-1 activated dimension of $x_m$ turns out to be the first one $(0.5)$. In other words, SAEs have a tendency to better interpret the relatively monosemantic features, whereas overlook the relatively polysemantic ones.

**Example 1** (Feature Shrinking). *Suppose $n_m = n = 3$, $n_p = 2$, $x = (0.5, 1.0, 0.8)^\top$, and $W_p = \begin{bmatrix} 1 & 0 & 0 \\ 0 & 1 & -1 \end{bmatrix}$. Then we have $x_p = (0.5, 0.2)^\top$ and*

$$x_m = \sigma(W_p^\top W_p x) = \sigma\left( \begin{bmatrix} 1 & 0 & 0 \\ 0 & 1 & -1 \\ 0 & -1 & 1 \end{bmatrix} \begin{bmatrix} 0.5 \\ \mathbf{1.0} \\ 0.8 \end{bmatrix} \right) = \begin{bmatrix} \mathbf{0.5} \\ 0.2 \\ 0 \end{bmatrix}. \tag{8}$$

**Feature Vanishing.** When feature shrinking is severe enough, some features in the polysemantic dimensions even vanish and can rarely recovered by an SAE. In Example 2, the second and third dimensions of the SAE-recovered feature $x_m$ vanish completely, making $x_m$ have even less effective dimensions than $x_p$.

**Example 2** (Feature Vanishing). *Suppose $n_m = n = 3$, $n_p = 2$, $x = (0.7, 0.5, 0.3)^\top$, and $W_p = \begin{bmatrix} 0 & \sqrt{3}/2 & -\sqrt{3}/2 \\ 1 & -1/2 & -1/2 \end{bmatrix}$. Then we have $x_p = W_p x = (0.1\sqrt{3}, 0.3)^\top$, but*

$$x_m = \sigma(W_p^\top W_p x) = \sigma\left( \begin{bmatrix} 1 & -1/2 & -1/2 \\ -1/2 & 1 & -1/2 \\ -1/2 & -1/2 & 1 \end{bmatrix} \begin{bmatrix} 0.7 \\ 0.5 \\ 0.3 \end{bmatrix} \right) = \begin{bmatrix} 0.3 \\ \mathbf{0} \\ \mathbf{0} \end{bmatrix}. \tag{9}$$

## 4.2 SAE Full Recovery under Extreme Sparsity

Despite the disappointing theoretical conclusion in Section 4.2, in this part, we propose one possible explanation for why SAEs do work well in some real-life cases, that is, the sparsity of the ground truth monosemantic features is the key to SAE feature recovery.

**Theorem 2** (Optimality under extreme sparsity). *Let $\mathcal{L}_{\mathrm{SAE}}$ be defined in equation 4. For $n_m \gneq n$, and the columns of $W_p$ have non-positive interferences, if $S \to 1$, then we have $W_m^* = I^*(W_p, \mathbf{0})^\top \in \arg\min_{W_m} \mathcal{L}_{\mathrm{SAE}}(W_m; x_p)$, where $I^*$ is formed by swapping row $i$ with row $j \neq i$ of the identity matrix. Accordingly, we have $I^* \sigma(W_m^* x_p) = x$ for arbitrary $x$.*

Theorem 2 shows that when the ground truth feature $x$ is extremely sparse and the hidden dimension $n_m$ is large enough, $W_p^\top$ is the optimal solution to SAEs in the sense of zero-padding and row-reordering. Note that in Theorem 2, the columns of $W_p$ are only required to have non-positive interferences, which follows from Elhage et al. (2022b) and is much weaker than the digon/polygon geometry condition required in Theorem 1. By such a solution, under the extreme sparsity condition $S \to 1$, the SAE-recovered features can fully recover the ground truth monosemantic feature $x$. Intuitively, under extreme sparsity, $x$ becomes 1-sparse with probability nearly 1, and feature shrinking and feature vanishing do not happen to 1-sparse features.

Moreover, in Theorem 3, we show that when $S \to 1$, the solution derived in Theorem 2 is unique.

**Theorem 3** (Uniqueness). *Let $\mathcal{L}_{\mathrm{SAE}}$ be defined in equation 4. For $n_m = n$ and the columns of $W_p$ have non-positive interferences, if $S \to 1$, then $W_m^* = I^* W_p^\top$ is the unique solution to $\arg\min_{W_m} \mathcal{L}_{\mathrm{SAE}}(W_m; x_p)$.*

## 4.3 A Reweighted Remedy for SAE Feature Recovery under General Sparsity

As we show in previous sections, SAEs fail to recover ground truth monosemantic features under general sparsity conditions. In this section, by taking a closer look, we show that this is largely

because the SAE loss is not a direct reconstruction of the ground truth monosemantic feature $x$. Instead, it reconstructs the polysemantic $x_p = W_p x$ because the ground truth $x$ is unknown. In this case, the superposition matrix $W_p$ could mistakenly match the reconstructed polysemantic feature $\tilde{x}_p = W_m^\top x_m$ with $x_p$ even if the reconstructed monosemantic features $x_m$ do not match the ground truth $x$. Specifically, in this section, we first identify a gap between the SAE loss and the reconstruction loss of ground truth $x$, and then we discuss how a reweighting strategy narrows the gap and enhances the ground truth reconstruction.

### 4.3.1 The Gap between SAE Reconstruction and Ground Truth Reconstruction

We first of all consider an ideal case, where we are allowed to directly reconstruct the ground truth monosemantic features $x$. Without loss of generality, we assume $n_m = n$, and define the ground truth reconstruction loss of $x$ as

$$\mathcal{L}_{\mathrm{GT}}(W_m; x) = \mathbb{E}_x \|x - x_m\|^2 = \mathbb{E}_x \|x - \sigma(W_m x_p)\|^2 = \mathbb{E}_x \|x - \sigma(W_m W_p x)\|^2. \quad (10)$$

Then by comparing $\mathcal{L}_{\mathrm{SAE}}$ and $\mathcal{L}_{\mathrm{GT}}$, we derive the theoretical gap between the two losses.

**Theorem 4** (Gap between $\mathcal{L}_{\mathrm{SAE}}$ and $\mathcal{L}_{\mathrm{GT}}$). *Let $\mathcal{L}_{\mathrm{SAE}}$ and $\mathcal{L}_{\mathrm{GT}}$ be defined in equation 4 and equation 10, respectively. Then when $W_m = W_p^\top$, we have*

$$\mathcal{L}_{\mathrm{SAE}}(W_m; x_p) - \mathcal{L}_{\mathrm{GT}}(W_m; x) = [x - \sigma(W_p^\top W_p x)]^\top (W_p^\top W_p - I_{n \times n})[x - \sigma(W_p^\top W_p x)]. \quad (11)$$

Theorem 4 shows that the gap between the SAE loss $\mathcal{L}_{\mathrm{SAE}}(W_m; x_p)$ and the ground truth reconstruction loss $\mathcal{L}_{\mathrm{GT}}(W_m; x)$. Note that according to Theorem 1, we consider the gap when the SAE loss reaches its optimal, i.e., $W_m = W_p^\top$. The gap term depends on two terms, i.e., the gap between ground truth $x$ and the recovered feature $x_m = \sigma(W_p^\top W_p x)$ and the gap between $W_p^\top W_p$ and the identity matrix $I_{n \times n}$. On the one hand, the gap term goes to zero if the features are perfectly recovered by an SAE. For instance, by Theorem 2, when $x$ is extremely sparse, we have $x = \sigma(W_p^\top W_p x)$ and therefore the gap vanishes. On the other hand, in general cases where SAEs fail to perfectly recover the ground truth $x$, i.e., $x \neq \sigma(W_p^\top W_p x)$, the gap term depends largely on $W_p^\top W_p - I_{n \times n}$, which unfortunately cannot be reduced because for SAEs, $W_p$ is given as an input rather than learned.

### 4.3.2 Adaptively ReWeighted Sparse AutoEncoders (WSAEs)

As one possible solution to narrow the gap shown in Theorem 4, we propose the *reWeighted Sparse AutoEncoders* (WSAEs) with adaptive weights according to the polysemantic level of each dimension. Specifically, given weights $\gamma_i > 0$, $i \in [n_p]$, we define the WSAE loss as

$$\mathcal{L}_{\mathrm{WSAE}}(W_m; x_p) = \mathbb{E}_{x_p} \|\Gamma[x_p - W_m^\top \mathrm{ReLU}(W_m x_p)]\|_2^2, \quad (12)$$

where $\Gamma = \mathrm{diag}(\gamma_1, \ldots, \gamma_{n_p})$.

**Theorem 5** (Gap between $\mathcal{L}_{\mathrm{WSAE}}$ and $\mathcal{L}_{\mathrm{GT}}$). *Let $\mathcal{L}_{\mathrm{WSAE}}$ and $\mathcal{L}_{\mathrm{GT}}$ be defined in equation 12 and equation 10, respectively. Then when $W_m = W_p^\top$, we have*

$$\mathcal{L}_{\mathrm{WSAE}}(W_m; x_p) - \mathcal{L}_{\mathrm{GT}}(W_m; x) = [x - \sigma(W_m W_p x)]^\top (W_p^\top \Gamma^\top \Gamma W_p - I_{n \times n})[x - \sigma(W_m W_p x)]. \quad (13)$$

Theorem 5 shows the gap between a reweighted SAE loss $\mathcal{L}_{\mathrm{SAE}}(W_m; x_p)$ and the reconstruction loss of the ground truth features $\mathcal{L}_{\mathrm{GT}}(W_m; x, x_p)$. Compared with Theorem 4 where $W_p^\top W_p - I_{n \times n}$ is fixed, when features are not perfectly recovered, i.e., $x \neq \sigma(W_m W_p x)$, the gap in Theorem 5 is adjustable w.r.t. the weight matrix $\Gamma$. That is, given a properly chosen $\Gamma$, we can effectively narrow the gap term in equation 13 and thus gain a better reconstruction of the ground truth features $x$. As a special case, when the weight matrix $\Gamma$ is an identity matrix (uniform weights), the gap of WSAEs in Theorem 5 degenerates to that of SAEs shown in Theorem 4.

Then we discuss how to select proper weights to narrow the gap term in equation 13. Specifically, we focus on the polysemantic level of each dimension. Observe that

$$W_p^\top \Gamma^\top \Gamma W_p - I_{n \times n} = \begin{bmatrix} \gamma_1^2 - 1 & \cdots & \gamma_1^2 W_{p,[:,1]}^\top W_{p,[:,n]} \\ \cdots & & \cdots \\ \gamma_n^2 W_{p,[:,n]}^\top W_{p,[:,1]} & \cdots & \gamma_n^2 - 1 \end{bmatrix}. \quad (14)$$

Then according to equation 14, for the relatively monosemantic dimensions with almost zero off-diagonal interference terms, we should assign weights near 1 to reduce $\gamma_i^2 - 1$, whereas for the relatively polysemantic dimensions, we should assign smaller weights to primarily reduce the off-diagonal negative interferences. In short, for WSAE reconstruction, we assign larger weights to the relatively monosemantic dimensions and smaller weights to the relatively polysemantic dimensions.

## 5 VALIDATION EXPERIMENTS

In this section, we conduct numerical experiments to validate our theoretical findings. In Section 5.1, we conduct validation experiments on synthetic data with known ground truth features. We validate that 1) SAEs in general fail to achieve full feature recovery unless the ground truth features are extremely sparse (Sections 4.1 and 4.2), and 2) our reweighting strategy can improve ground truth reconstruction when the sparsity level is low (Section 4.3). In Section 5.2, we conduct real-data experiments on both pretrained language and vision models to validate the effectiveness of our reweighted strategy in enhancing feature monosemanticity.

### 5.1 VALIDATION EXPERIMENTS THROUGH SYNTHETIC DATA

**Data Generation.** We follow the toy model settings in Elhage et al. (2022b). Specifically, the polysemantic features are generated by equation 1 described in Section 3, where the superposition matrix $W_p$ is learned by reconstructing the ground truth $x$ with a ReLU output model. We set the ground truth monosemantic feature dimension $n = 200$ and input polysemantic feature dimension $n_p = 20$ to generate polysemantic embeddings and then attempt to extract monosemantic features with SAEs trained on frozen polysemantic embeddings. When analyzing the influence of input sparsity, we set $S \in [0, 1]$ the sparse factor, and let $x_i = 0$ with a probability $S$, and $x_i \sim \mathcal{U}(0, 1)$ with probability $1 - S$, for $i \in [n]$.

**Setups.** For both SAE and WSAE, we adopt ReLU activation function, and set the hidden dimension as $n_m = n = 200$ (10 times the input dimension). For the weight selection in WSAE, we treat $W_p$ as unknown, and use the per-dimensional variance of $x_p$ as a proxy for monosemanticity. Specifically, let $s_i$ denote the variance of $x_p$ in the $i$-th dimension. Then given a tunable parameter $\alpha > 0$, we set the weights as $\gamma_i = s_i^\alpha$. As we find that the experimental results are relatively robust against the choice of $\alpha$, we set $\alpha = 1$ for the experiments.

**Validation of SAE feature recovery.** We validate the theoretical results in Sections 4.1 and 4.2 that SAEs in general fail to achieve full feature recovery unless the ground truth features are extremely sparse. Specifically, under different sparsity levels $S \in [0, 1]$, we measure the monosemanticity of SAE latents by the average number of ground-truth features activated in the same SAE dimension (with more details shown in Appendix C.3). If the SAE successfully recovers monosemantic features, each latent should be activated by only a small number of input features. As shown in Figure 2, we observe a clear decrease in the average number of activated features (indicating a significant improvement in the monosemanticity) with increased input sparsity. This empirical result aligns well with our theoretical findings that SAEs fully recover ground truth monosemantic features only when the sparsity level is extreme.

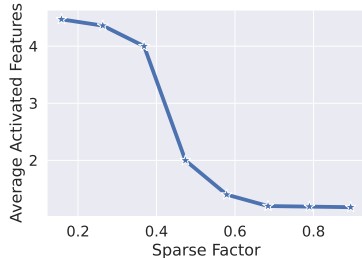

Figure 2: Monosemanticity (measured by the average activated features) of SAE features increases with increasing sparsity of ground truth monosemantic features.

**Validation of WSAE ground truth reconstruction.** We validate the theoretical discussions in Section 4.3 that WSAE features have better ground truth reconstruction than SAE features when the sparsity level of ground truth features is low. Specifically, we respectively measure the reconstruction loss of ground truth monosemantic features $\mathcal{L}_{GT}$ defined in equation 10, the reconstruction loss of polysemantic features $\mathcal{L}_{\text{SAE}}$ and $\mathcal{L}_{\text{WSAE}}$ defined in equation 4 and equation 12, and the monosemanticity level measured by per-dimensional variance (where larger per-dimensional variance indicates higher monosemanticity), respectively. In Figure 3, when the sparsity level of the ground truth features is relatively low, we show that the WSAE-recovered features have better reconstruction of the ground truth monosemantic features $x$ compared with SAE-recovered features, whereas maintaining

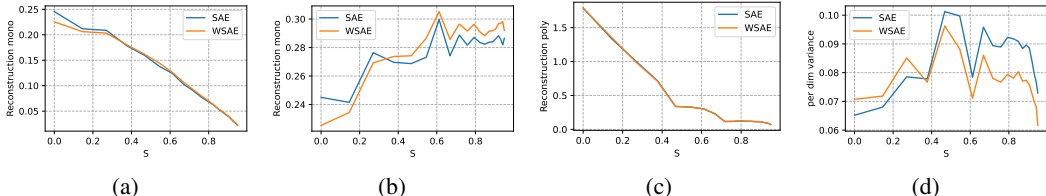

Figure 3: Validation experiments of WSAE ground truth reconstruction on synthetic data. (a) Ground truth reconstruction error $\mathcal{L}_{\mathrm{GT}}$, where WSAE has lower error compared with SAE when the sparsity level $S$ is low. (b) Reconstruction error on the non-sparse dimensions of the ground truth monosemantic features, showing a greater error gap between WSAE and SAE. (c) The reconstruction error of the polysemantic features $x_p$, where the errors of the two methods are comparable. (d) Monosemanticity measured by per-dimensional variance, where WSAE features are more monosemantic compared with SAE features when the sparsity level is low.

a comparable reconstruction of the polysemantic features $x_p$. Specifically, WSAE has a lower ground truth reconstruction error compared to SAE (Figure 3(a)), and if we evaluate only on the non-sparse dimensions of the ground truth $x$, the advantage of WSAE over SAE under low sparsity becomes more significant (Figure 3(b)). These empirical findings well verify the theoretical discussions in Section 4.3 that a proper weight selection can narrow the gap between WSAE reconstruction error and ground truth reconstruction error. In addition, in Figure 3(c), we show that the SAE and WSAE reconstruction errors of the polysemantic features $x_p$ are comparable, indicating that WSAEs do not lie far away from the Pareto-frontier of sparsity and reconstruction. In Figure 3(d), we show that when sparsity is relatively low, WSAE features have better monosemanticity than SAE features, indicating better recovery of the ground truth monosemantic features.

## 5.2 Empirical Evaluation of Reweighted SAEs through Real Data

In this section, we evaluate the effectiveness of our proposed strategies on real-world datasets. We evaluate on both pretrained language models and vision models.

### 5.2.1 Experiments on Language Models

**Setups.** We use Pythia-160M (Biderman et al., 2023) as the frozen language model backbone and train sparse autoencoders (SAEs) on its internal activations. The latent dimension of the SAE is set to 32 times the input dimension, and we employ Top-$k$ activation ($k = 32$) in the latent layer of SAEs. Guided by the theoretical analysis, we compare two training objectives: a standard reconstruction loss with uniform weights, and a weighted reconstruction loss that emphasizes monosemantic features.

In real-world settings, we cannot directly access the ground-truth monosemanticity of features. Nonetheless, several surrogate metrics have been proposed to evaluate monosemanticity. For instance, Wang et al. (2024a) uses variance within each feature dimension as a surrogate while Paulo et al. (2024) introduces the auto-interpretability score, which leverages large language models to summarize whether the features activated in the same dimension are semantically similar. In our experiments, we use the variance as the metric when assigning weights, due to its computational simplicity. As discussed in Wang et al. (2024a), monosemantic neurons have high deviation of activation value and low frequency of activation, so higher variance indicates stronger monosemanticity. Specifically, we let $s_i$ denote the variance of activations in the SAE latent dimension $i$. We set the weights as $\gamma_i = s_i^\alpha$ for $i \in [n_p]$, where $\alpha > 0$ is a tunable parameter. When $\alpha$ increases, the reconstruction loss assigns more weights to the reconstruction of monosemantic features.

On the other hand, for the evaluation of SAE latents, we use the auto-interpretability score (Paulo et al., 2024) as it is more accurate to assess monosemanticity. To compute the score for a given SAE latent dimension, we first identify the top-activated samples for that dimension. A large language model (e.g., Llama3.1-8B Touvron et al. (2023)) is then used to generate a natural language summary describing the shared characteristics of these samples. In the next step, another language model is prompted with the generated summary to predict whether additional samples in the dataset would activate the given SAE dimension. The prediction accuracy serves as the auto-interpretability

Table 1: Auto-interpretability scores (%) of SAEs trained following different layers (0-11) of Pythia-160M with original SAE and weighted SAE loss. SAEs trained with weighted SAE loss obtain higher auto-interpretability scores (i.e., stronger monosemanticity) across different situations.

| | 0 | 1 | 2 | 3 | 4 | 5 | 6 | 7 | 8 | 9 | 10 | 11 |
|---|---|---|---|---|---|---|---|---|---|---|---|---|
| Original SAE | 74.7 | 74.1 | 76.7 | 77.8 | 78.5 | 79.5 | 79.3 | 77.8 | 74.6 | 75.6 | 71.6 | 72.5 |
| Weighted SAE ($\alpha$=0.5) | 75.4 | 77.6 | 76.4 | 77.9 | 79.3 | 79.6 | 79.8 | 77.4 | 78.6 | **79.3** | **76.1** | 72.6 |
| Gains | +0.7 | +3.5 | -0.3 | +0.1 | +0.8 | +0.1 | +0.5 | -0.4 | +4.0 | +3.7 | +4.5 | +0.1 |
| Weighted SAE ($\alpha$=1) | **77.2** | **78.9** | **81.3** | **84.6** | **83.9** | **83.3** | **83.9** | **79.6** | **81.5** | 77.6 | 72.4 | **73.5** |
| Gains | +2.5 | +4.8 | +4.6 | +6.8 | +5.4 | +3.8 | +4.6 | +1.8 | +6.9 | +2.0 | +0.8 | +1.0 |

score: higher accuracy indicates that the activated samples in the dimension are semantically similar, reflecting stronger monosemanticity. More details can be found in Appendix C.

**Results.** As shown in Table 1, assigning greater weight to the reconstruction of monosemantic features leads to a significant improvement in the monosemanticity of SAE latents, yielding an average 3.8% gain in auto-interpretability score when $\alpha = 1$. The improvements are consistent across SAEs trained on different layers of backbone models. We also note that the improvement is more significant when the baseline SAE already exhibits relatively strong monosemanticity. These empirical findings further support our theoretical insight in Section 4.3 that reweighting SAE with monosemanticity level can enhance the recovery of the monosemantic features. As a supplement, additional experiments with an alternative Top-$k$ realization, evaluations on the Llama model, and sensitivity analysis are shown in Appendix C.1.

### 5.2.2 EXPERIMENTS ON VISION MODELS

In addition to language models, we also evaluate our strategy on SAEs trained following pretrained vision backbones. As noted by Wang et al. (2024b), features learned by current vision pertaining paradigms, such as contrastive learning, are often highly polysemantic, making it difficult to select monosemantic dimensions. Consequently, we adopt Non-negative Contrastive Learning (NCL) (Wang et al., 2024b) to pretrain the backbone, as it can simultaneously obtain monosemantic and polysemantic features.

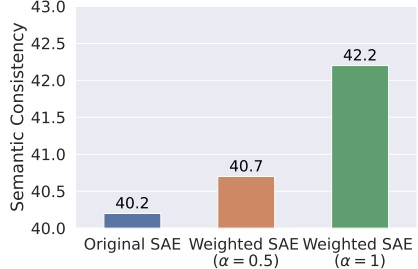

Figure 4: Semantic consistency (%) of SAEs trained on the embeddings of ResNet-18 with original SAE and weighted SAE loss.

Specifically, we pretrain a ResNet-18 with NCL on ImageNet-100, and subsequently train SAEs on the learned representations. During the training process of SAEs, we set the latent dimension to 16384 and use the topK sparse autoencoder with $k = 16$. When evaluating monosemanticity, we follow Wang et al. (2024b) and use semantic consistency as the metric, which calculates the proportion of top-activated samples that belong to their most frequent class along each dimension. As the higher semantic consistency indicates stronger monosemanticity, we respectively train SAEs with uniform weights and more weights on the dimension with higher semantic consistency. Specifically, denote $\beta_i$ as the semantic consistency of the $i$-th dimension, and we set the weights as $\gamma_i = \beta_i^{\alpha}$ for $i \in [n_p]$, where $\alpha > 0$ is a tunable parameter. In Figure 4, we observe that assigning greater weight to the reconstruction of monosemantic features leads to a notable improvement in semantic consistency (monosemanticity) of SAE latents. These results further validate both our theoretical analysis and the effectiveness of the proposed weighting strategy.

## 6 CONCLUSION

In this paper, we establish a theoretical framework to reveal the inherent limits of SAEs to recover ground-truth monosemantic features. Specifically, based on a closed-form theoretical solution, we show that standard SAEs inevitably suffer from feature shrinking and vanishing, preventing full recovery except when the true features are extremely sparse. While many interpretability studies

assume that increasing sparsity or width of SAEs improves feature separation indefinitely, this work shows that such improvement plateaus due to intrinsic representational interference, meaning full disentanglement is mathematically impossible under realistic sparsity. Consequently, SAE-based interpretability should be regarded as an approximation tool, not as a faithful feature recovery mechanism. This reframes SAE-derived neurons as approximate projections of overlapping features, rather than direct encodings of ground-truth concepts.

Furthermore, to enhance feature recovery in the low sparsity situations, we introduced a simple yet effective strategy called reWeighted Sparse Autoencoder (WSAE) and propose a theoretical weight-selection rule. Validation experiments on both synthetic and real data confirm our theory and demonstrate that WSAEs can enhance monosemanticity and interpretability of features learned. We note that the proposed WSAE serves as just one exemplar remedy, and our theoretical framework has the potential to motivate further methodological advances (e.g. alternative matrix designs to close the loss gap) aimed at overcoming the fundamental feature-recovery limits of SAEs.

## ACKNOWLEDGEMENT

Yisen Wang was supported by National Natural Science Foundation of China (92370129, 62376010), Beijing Major Science and Technology Project under Contract no. Z251100008425006, Beijing Nova Program (20230484344, 20240484642), and State Key Laboratory of General Artificial Intelligence.

## ETHICS STATEMENT

This work makes use of publicly available datasets and models. No private or sensitive data are involved, and no harmful content is included. Therefore, we believe this paper does not raise any ethical concerns.

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

## A PROOFS

Without loss of generality, we consider the ReLU sparse activation function for proofs, i.e. $\sigma(x) = \text{ReLU}(x) = x\mathbf{1}[x > 0]$. Note that for other activation functions such as TopK or Jump-ReLU, the theoretical results are similar because they are based on the ReLU function form. Specifically, we have Jump-ReLU$(x, c) = \text{ReLU}(x - c)$ and Top-$k(x) = \text{ReLU}(x - x_{(k+1)}\mathbf{1}) + x_{(k+1)}\text{H}(x - x_{(k+1)}\mathbf{1})$, where $H(x) := \mathbf{1}[x \geq 0]$ denotes the Heaviside function, $x_{(k+1)}$ denotes the $(k + 1)$-th largest element of $x$, and $\mathbf{1}$ denotes the all-one vector. The sparse activation functions have a general form $\sigma(x) = \text{ReLU}(x - c) + b$ and the general gradient form $\frac{\partial \sigma(x)}{\partial x} = \text{H}(x - c)$.

Also note that when $I^*$ is formed by swapping row $i$ with row $j \neq i$ of the identity matrix $I$, if $W_p^\top$ is a solution to the SAE loss minimization, then $I^* W_p^\top$ is also a solution because

$$
\begin{aligned}
\mathcal{L}_{\text{SAE}}(I^* W_p^\top; x_p) &= \mathbb{E}_x \|W_p x - (I^* W_p^\top)^\top \sigma(I^* W_p^\top W_p x)\|^2 \\
&= \mathbb{E}_x \|W_p x - W_p (I^*)^\top I^* \sigma(W_p^\top W_p x)\|^2 \\
&= \mathbb{E}_x \|W_p x - W_p \sigma(W_p^\top W_p x)\|^2 \\
&= \mathcal{L}_{\text{SAE}}(W_p^\top; x_p),
\end{aligned}
\tag{15}
$$

where the second last equation holds because $(I^*)^\top I^* = I$. In the following, we will not repetitively prove this point in the proofs of Theorem 1, 2, and 3.

### A.1 PROOFS RELATED TO SECTION 4.1

*Proof of Theorem 1.* By definition, we have

$$
\begin{aligned}
&\mathcal{L}_{\text{SAE}}(W_m; x_p) \\
&= \mathbb{E}_x \sum_{i \in [n_p]} [W_{p,[i,:]} x - W_{m,[:,i]}^\top \sigma(W_m W_p x)]^2 \\
&= \mathbb{E}_x \sum_{i \in [n_p]} [W_{p,[i,:]} x - \sum_{k \in [n]} W_{m,[k,i]}^\top \sigma(\sum_{j \in [n_p]} W_{m,[k,j]} W_{p,[j,:]} x)]^2.
\end{aligned}
\tag{16}
$$

For $u \in [n]$, $v \in [n_p]$, we have

$$
\begin{aligned}
\frac{\partial \mathcal{L}_{\text{SAE}}}{\partial W_{m,[u,v]}} = \mathbb{E}_x \Big\{ &- 2[W_{p,[v,:]} x - W_{m,[:,v]}^\top \sigma(W_m W_p x)] \sigma(W_{m,[u,:]} W_p x) \\
&- 2 \sum_{i \in [n_p]} [W_{p,[i,:]} x - W_{m,[:,i]}^\top \sigma(W_m W_p x)] W_{m,[u,i]} H(W_{m,[u,:]} W_p x) W_{p,[v,:]} x \Big\},
\end{aligned}
\tag{17}
$$

where $H(\cdot)$ denotes the Heaviside function. Observe that for non-zero $x$, $\frac{\partial \mathcal{L}_{\text{SAE}}}{\partial W_{m,[u,v]}} = 0$ if and only if

$$
\mathbb{E}_x [W_{p,[i,:]} x - W_{m,[:,i]}^\top \sigma(W_m W_p x)] = 0.
\tag{18}
$$

When $W_m = W_p^\top$, because the columns of $W_p$ form digons/polygons, we have $W_p^\top W_p$ being a block diagonal matrix, with block $B_1 = I_{m \times m}$ for the $m$ ($0 \leq m \leq n_p$) monosemantic dimensions and blocks

$$
B_j = \begin{bmatrix} 1 & -1/(p_j - 1) & \cdots & -1/(p_j - 1) \\ & \cdots & & \\ -1/(p_j - 1) & -1/(p_j - 1) & \cdots & 1 \end{bmatrix}
\tag{19}
$$

for the polysemantic dimensions, where $p_j$ is the number of ground truth monosemantic features entangled in the $j$-th block. Then we have

$$
\sigma(W_m W_p x) = \sigma\left( \begin{bmatrix} I_{m \times m} & 0 & \cdots & 0 \\ 0 & B_2 & \cdots & 0 \\ & & \cdots & \\ 0 & 0 & \cdots & B_b \end{bmatrix} x \right) = \begin{bmatrix} \sigma(x_{[1:m]}) \\ \sigma(B_2 x_{[m+1:m+p_j]}) \\ \cdots \\ \sigma(B_b x_{[n-p_b+1:n]}) \end{bmatrix}.
\tag{20}
$$

Denote $P_j = m + \sum_{1 < l \leq j} p_l$ and note that

$$\mathbb{E}_x \sigma(B_j x_{[P_{j-1}+1:P_j]}) = \begin{bmatrix} \mathbb{E}_x \sigma(x_{P_{j-1}+1} - \frac{1}{p_j-1} \sum_{P_{j-1}+1 < l \leq P_j} x_l) \\ \cdots \\ \mathbb{E}_x \sigma(x_{P_j} - \frac{1}{p_j-1} \sum_{P_{j-1}+1 \leq l < P_j} x_l) \end{bmatrix}. \tag{21}$$

For $i \in [p_j]$, we have

$$\mathbb{E}_x \sigma\Big(x_{P_{j-1}+i} - \frac{1}{p_j-1} \sum_{P_{j-1}+1 \leq l \leq P_j, l \neq i} x_l\Big)$$

$$= \mathbb{E}_x \Big(x_{P_{j-1}+i} - \frac{1}{p_j-1} \sum_{P_{j-1}+1 \leq l \leq P_j, l \neq i} x_l \Big| x_{P_{j-1}+i} > \frac{1}{p_j-1} \sum_{P_{j-1}+1 \leq l \leq P_j, l \neq i} x_l\Big)$$

$$\cdot \mathrm{P}\Big(x_{P_{j-1}+i} > \frac{1}{p_j-1} \sum_{P_{j-1}+1 \leq l \leq P_j, l \neq i} x_l\Big)$$

$$:= \mu_{P_{j-1}+i}. \tag{22}$$

Because $x_i$'s are i.i.d., we have $\mu_{P_{j-1}+1} = \cdots = \mu_{P_{j-1}+p_j} := \mu_j$. Denote $\mu = \mathbb{E}_x x_i$, $\mathbf{1} = (1, \ldots, 1)^\top$ as the all-one vector, and $\mathbf{0} = (0, \ldots, 0)^\top$ as the all-zero vector. Then we have

$$\mathbb{E}_x W_p \sigma(B_j x_{[P_{j-1}+1:P_j]}) = W_p \mathbb{E}_x \sigma(B_j x_{[P_{j-1}+1:P_j]}) = W_p \begin{bmatrix} \mu \mathbf{1}_m \\ \mu_2 \mathbf{1}_{p_2} \\ \cdots \\ \mu_b \mathbf{1}_{p_b} \end{bmatrix} = \begin{bmatrix} \mu \mathbf{1}_m \\ \mu_2 \mathbf{0}_{p_2} \\ \cdots \\ \mu_b \mathbf{0}_{p_b} \end{bmatrix} = \begin{bmatrix} \mu \mathbf{1}_m \\ \mathbf{0}_{n-m} \end{bmatrix}, \tag{23}$$

where the second last equation holds because the columns of $W_p$ form digons/polygons.

Therefore, when $W_m = W_p^\top$, we have equation 18 holds because

$$W_p \mathbb{E}_x x - W_p \sigma(W_m W_p x) = \mu W_p \mathbf{1} - \begin{bmatrix} \mu \mathbf{1}_m \\ \mu_j \mathbf{0}_{n-m} \end{bmatrix} = \mu \begin{bmatrix} \mathbf{1}_m \\ \mathbf{0}_{n-m} \end{bmatrix} - \begin{bmatrix} \mu \mathbf{1}_m \\ \mathbf{0}_{n-m} \end{bmatrix} = 0, \tag{24}$$

i.e., $W_p^\top \in \arg\min_{W_m} \mathcal{L}_{\mathrm{SAE}}(W_m; x_p)$. $\qquad\square$

## A.2 Proofs Related to Section 4.2

*Proof of Theorem 2.* When $W_m = (W_p, \mathbf{0})^\top$, we have

$$\mathcal{L}_{\mathrm{SAE}}(W_m; x_p) = \mathbb{E}_x \|W_p x - (W_p, \mathbf{0})\sigma(\begin{bmatrix} W_p^\top \\ \mathbf{0} \end{bmatrix} W_p x)\|^2$$

$$= \mathbb{E}_x \|W_p x - W_p^\top \sigma(W_p^\top W_p x)\|^2. \tag{25}$$

When $S \to 1$, we have

$$\mathcal{L}_{\mathrm{SAE}}(W_m; x_p)$$

$$= \sum_{u=1}^n S^n \mathcal{L}(W_m; W_p, x = \mathbf{0}) + \sum_{u=1}^n \mathbb{E}_{x_u} C_n^1 S^{n-1}(1-S)\mathcal{L}(W_m; W_p, x = x_u e_u) + o(1-S)$$

$$= 0 + n S^{n-1}(1-S) \sum_{u=1}^n \mathbb{E}_{x_u} \mathcal{L}(W_m; W_p, x = x_u e_u) + o(1-S)$$

$$= n S^{n-1}(1-S) \sum_{u=1}^n \mathbb{E}_{x_u} \mathcal{L}(W_m; W_p, x = x_u e_u)$$

$$= n S^{n-1}(1-S) \sum_{u=1}^n \mathbb{E}_{x_u} \|W_{p,[:,u]} - W_p^\top \sigma(W_p^\top W_{p,[:,u]})\|^2. \tag{26}$$

When the superposed dimensions have negative interferences as discussed in Elhage et al. (2022b), i.e., $W_{p,[:,i]}^\top W_{p,[:,j]} \leq 0$ for $i \neq j$, we have $\mathcal{L}_{\mathrm{SAE}}(W_m; x_p) = 0 = \min_{W_m} \mathcal{L}_{\mathrm{SAE}}(W_m; x_p)$.

$\qquad\square$

Also note that for TopK or JumpReLU activations, feature recovery under potential positive interference is also possible. For example, for top-$k$ activation, when $1 < k \leq n_m - p$, where $p := \max_{i \in [n]} |\{j \in [n] : W_{p,[:,i]}^\top W_{p,[:,j]} < 0\}|$, we have Top-$k(W_p^\top W_{p,[:,u]}) = e_u$, and correspondingly $\mathcal{L}_{\mathrm{SAE}}(W_m; x_p) = 0 = \min_{W_m} \mathcal{L}_{\mathrm{SAE}}(W_m; x_p)$.

*Proof of Theorem 3.* Then by definition, we have

$$
\begin{aligned}
&\mathcal{L}_{\mathrm{SAE}}(W_m; x_p) \\
&= \mathbb{E}_x \| W_p x - W_m^\top \sigma(W_m W_p x) \|_2^2 \\
&= \mathbb{E}_x [W_p x - W_m^\top \sigma(W_m W_p x)]^\top [W_p x - W_m^\top \sigma(W_m W_p x)] \\
&= \mathbb{E}_x x^\top W_p^\top W_p x - 2 x^\top W_p^\top W_m^\top \sigma(W_m W_p x) \\
&\quad + \sigma(W_m W_p x)^\top W_m W_m^\top \sigma(W_m W_p x) \\
&= \mathbb{E}_x x^\top W_p^\top W_p x - 2 \sum_{q=1}^{n_p} \sum_{j=1}^{n_m} x^\top W_{p,[q,:]}^\top W_{m,[j,q]} \sigma(\sum_{i=1}^{n_p} W_{m,[j,i]} W_{p,[i,:]} x) \\
&\quad + \sum_{j=1}^{n_m} \sum_{q=1}^{n_p} \sigma(\sum_{i=1}^{n_p} W_{m,[j,i]} W_{p,[i,:]} x) W_{m,[j,q]} \sum_{s=1}^{n_m} W_{m,[s,q]} \sigma(\sum_{t=1}^{n_p} W_{m,[s,t]} W_{p,[t,:]} x).
\end{aligned} \tag{27}
$$

If $S \to 1$, by equation 26, we have

$$
\mathcal{L}_{\mathrm{SAE}}(W_m; x_p) = n S^{n-1}(1 - S) \sum_{u=1}^{n} \mathbb{E}_{x_u} \mathcal{L}_u(W_m; W_p, x_u), \tag{28}
$$

where for $u \in [n]$ and $x_u > 0$, by equation 27, we denote

$$
\begin{aligned}
&\mathcal{L}_u(W_m; W_p, x_u) \\
&:= x_u^2 W_{p,[:,u]}^\top W_{p,[:,u]} - 2 x_u^2 \sum_{q=1}^{n_p} \sum_{j=1}^{n_m} W_{p,[q,u]} W_{m,[j,q]} \sigma(\sum_{i=1}^{n_p} W_{m,[j,i]} W_{p,[i,u]}) \\
&\quad + x_u^2 \sum_{j=1}^{n_m} \sum_{q=1}^{n_p} \sigma(\sum_{i=1}^{n_p} W_{m,[j,i]} W_{p,[i,u]}) W_{m,[j,q]} \sum_{s=1}^{n_m} W_{m,[s,q]} \sigma(\sum_{t=1}^{n_p} W_{m,[s,t]} W_{p,[t,u]}).
\end{aligned} \tag{29}
$$

Then for $k \in [n_m]$ and $l \in [n_p]$, by equation 26, we have

$$
\begin{aligned}
&\frac{\partial \mathcal{L}_u(W_m; W_p, x_u)}{\partial W_{m,[k,l]}} \\
&= -2 x_u^2 W_{p,[l,u]} \sigma(\sum_{i=1}^{n_p} W_{m,[k,i]} W_{p,[i,u]}) \\
&\quad - 2 x_u^2 \sum_{q=1}^{n_p} W_{p,[q,u]} W_{m,[k,q]} \mathrm{H}(\sum_{i=1}^{n_p} W_{m,[k,i]} W_{p,[i,u]}) W_{p,[l,u]} \\
&\quad + x_u^2 \sum_{q=1}^{n_p} \mathrm{H}(\sum_{i=1}^{n_p} W_{m,[k,i]} W_{p,[i,u]}) W_{p,[l,u]} W_{m,[k,q]} \sum_{s=1}^{n_m} W_{m,[s,q]} \sigma(\sum_{t=1}^{n_p} W_{m,[s,t]} W_{p,[t,u]}) \\
&\quad + x_u^2 \sigma(\sum_{i=1}^{n_p} W_{m,[k,i]} W_{p,[i,u]}) \sum_{s=1}^{n_m} W_{m,[s,l]} \sigma(\sum_{t=1}^{n_p} W_{m,[s,t]} W_{p,[t,u]}) \\
&\quad + x_u^2 \sum_{j=1}^{n_m} \sigma(\sum_{i=1}^{n_p} W_{m,[j,i]} W_{p,[i,u]}) W_{m,[j,l]} \sigma(\sum_{t=1}^{n_p} W_{m,[k,t]} W_{p,[t,u]}) \\
&\quad + x_u^2 \sum_{j=1}^{n_m} \sum_{q=1}^{n_p} \sigma(\sum_{i=1}^{n_p} W_{m,[j,i]} W_{p,[i,u]}) W_{m,[j,q]} W_{m,[k,q]} \mathrm{H}(\sum_{t=1}^{n_p} W_{m,[k,t]} W_{p,[t,u]}) W_{p,[l,u]}
\end{aligned}
$$

$$= 2x_u^2 \sum_{q=1}^{n_p} \mathrm{H}(\sum_{i=1}^{n_p} W_{m,[k,i]} W_{p,[i,u]}) W_{p,[l,u]} W_{m,[k,q]}$$

$$\cdot \Big[ \sum_{s=1}^{n_m} W_{m,[s,q]} \sigma(\sum_{t=1}^{n_p} W_{m,[s,t]} W_{p,[t,u]}) - W_{p,[q,u]} \Big]$$

$$+ 2x_u^2 \sigma(\sum_{i=1}^{n_p} W_{m,[k,i]} W_{p,[i,u]}) \Big[ \sum_{s=1}^{n_m} W_{m,[s,l]} \sigma(\sum_{t=1}^{n_p} W_{m,[s,t]} W_{p,[t,u]}) - W_{p,[l,u]} \Big]. \tag{30}$$

Observe that $\frac{\partial \mathcal{L}_u(W_m; W_p, x_u)}{\partial W_{m,[k,l]}} = 0$ for $k \in [n_m]$ and $l \in [n_p]$ holds if and only if

$$\sum_{s=1}^{n_m} W_{m,[s,l]} \sigma(\sum_{t=1}^{n_p} W_{m,[s,t]} W_{p,[t,u]}) = W_{p,[l,u]} \tag{31}$$

for $l \in [n_p]$, or equivalently

$$W_{p,[:,u]} = W_m^\top \sigma(W_m W_{p,[:,u]}) \tag{32}$$

$$W_p = W_m^\top \sigma(W_m W_p). \tag{33}$$

When $n_m = n$ and the superposed dimensions have negative interferences as discussed in Elhage et al. (2022b), i.e., $W_{p,[:,i]}^\top W_{p,[:,j]} \le 0$ for $i \ne j$, for arbitrary $W_p \in \mathbb{R}^{n_p \times n}$, the unique solution is $W_m = W_p^\top$ (see the following for details).

For a special example, if $W_p = I_{2 \times 2}$, equation 33 becomes $I = W_m^\top \sigma(W_m)$, and we observe that $W_m = W_p^\top = I$ is a solution. Then if $W_m \ne I_{2 \times 2}$, we can have a decomposition that

$$W_m = I + E_1 - E_2 + D, \tag{34}$$

where $E_1$ and $E_2$ are off-diagonal with non-negative elements on mutually exclusive positions, and $D$ is a diagonal matrix. Then equation 33 becomes

$$I = (I + E_1 - E_2 + D)^\top \sigma(I + E_1 - E_2 + D)$$
$$= (I + E_1 - E_2 + D)^\top \sigma(I + E_1 + D)$$
$$= I + E_1^\top - E_2^\top + D^\top + E_1 + E_1^\top E_1 - E_2^\top E_1 + D^\top E_1 + D + E_1^\top D - E_2^\top D + D^\top D$$
$$= I + (2D + D^2 + E_1^\top E_1) + (E_1^\top - E_2^\top + E_1), \tag{35}$$

where the last equation holds because $E_2^\top E_1 = D^\top E_1 = E_2^\top D = 0$. Note that $2D + D^2 + E_1^\top E_1$ is diagonal and $E_1^\top - E_2^\top + E_1$ is off-diagonal, so there has to hold that $2D + D^2 + E_1^\top E_1 = 0$ and $E_2 = E_1 + E_1^\top$. Let $E_1 = \begin{bmatrix} 0 & \varepsilon_1 \\ \varepsilon_2 & 0 \end{bmatrix}$ with $\varepsilon_1, \varepsilon_2 \ge 0$. Then there has to hold $E_2 = E_1 + E_1^\top = \begin{bmatrix} 0 & \varepsilon_1 + \varepsilon_2 \\ \varepsilon_1 + \varepsilon_2 & 0 \end{bmatrix}$, which contradicts with that $E_1$ and $E_2$ have non-negative elements on mutually exclusive positions if $\varepsilon_1$ or $\varepsilon_2 > 0$. Therefore, it has to be $E_1 = E_2 = 0$, and accordingly $D = 0$. That is, we prove that $W_m = I$ is the unique solution to $I = W_m^\top \sigma(W_m)$, and that $W_m = W_p^\top$ is the unique solution to equation 33 for all $W_p$.

$\square$

## A.3 Proofs Related to Section 4.3

*Proof of Theorem 4.* By definition, we have

$$\mathcal{L}_{\mathrm{SAE}}(W_m; x_p) = \mathbb{E}_x \| W_p x - W_m^\top \sigma(W_m W_p x) \|^2$$
$$= \mathbb{E}_x \| W_p[x - \sigma(W_m W_p x)] + (W_p - W_m^\top) \sigma(W_m W_p x) \|^2$$
$$= \mathbb{E}_x \| W_p[x - \sigma(W_m W_p x)] \|^2 + \| (W_m - W_p^\top)^\top \sigma(W_m W_p x) \|^2$$
$$+ 2\sigma(W_m W_p x)^\top (W_m - W_p^\top) W_p[x - \sigma(W_m W_p x)]. \tag{36}$$

The first term of equation 36 can be further decomposed into

$$
\begin{aligned}
\|W_p[x &- \sigma(W_m W_p x)]\|^2 \\
&= [x - \sigma(W_m W_p x)]^\top W_p^\top W_p [x - \sigma(W_m W_p x)] \\
&= [x - \sigma(W_m W_p x)]^\top [x - \sigma(W_m W_p x)] \\
&\quad + [x - \sigma(W_m W_p x)]^\top (W_p^\top W_p - I_{n \times n})[x - \sigma(W_m W_p x)] \\
&= \|x - \sigma(W_m W_p x)\|^2 \\
&\quad + [x - \sigma(W_m W_p x)]^\top (W_p^\top W_p - I_{n \times n})[x - \sigma(W_m W_p x)]. \tag{37}
\end{aligned}
$$

Combining equation 36, equation 37, and that $W_m - W_p^\top = 0$ finishes the proof. $\square$

*Proof of Theorem 5.* By definition, we have

$$
\begin{aligned}
\mathcal{L}_{\text{WSAE}}(W_m; x_p) &= \mathbb{E}_x \|\Gamma[W_p x - W_m^\top \sigma(W_m W_p x)]\|^2 \\
&= \mathbb{E}_x \|\Gamma W_p[x - \sigma(W_m W_p x)] + \Gamma(W_p - W_m^\top)\sigma(W_m W_p x)\|^2 \\
&= \mathbb{E}_x \|\Gamma W_p[x - \sigma(W_m W_p x)]\|^2 + \|\Gamma(W_m - W_p^\top)^\top \sigma(W_m W_p x)\|^2 \\
&\quad + 2\sigma(W_m W_p x)^\top (W_m - W_p^\top)\Gamma^\top \Gamma W_p[x - \sigma(W_m W_p x)]. \tag{38}
\end{aligned}
$$

The first term of equation 38 can be further decomposed into

$$
\begin{aligned}
\|\Gamma W_p[x &- \sigma(W_m W_p x)]\|^2 \\
&= [x - \sigma(W_m W_p x)]^\top W_p^\top \Gamma^\top \Gamma W_p [x - \sigma(W_m W_p x)] \\
&= [x - \sigma(W_m W_p x)]^\top [x - \sigma(W_m W_p x)] \\
&\quad + [x - \sigma(W_m W_p x)]^\top (W_p^\top \Gamma^\top \Gamma W_p - I_{n \times n})[x - \sigma(W_m W_p x)] \\
&= \|x - \sigma(W_m W_p x)\|^2 \\
&\quad + [x - \sigma(W_m W_p x)]^\top (W_p^\top \Gamma^\top \Gamma W_p - I_{n \times n})[x - \sigma(W_m W_p x)]. \tag{39}
\end{aligned}
$$

Combining equation 38, equation 39, and that $W_m - W_p^\top = 0$ finishes the proof. $\square$

## B  DISCUSSIONS OF THE THEORETICAL FRAMEWORK

In this section, we discuss the significance and potential extensions of our theoretical results.

**Relationship to Sparse dictionary learning (SDL) and identifiability.** Sparse dictionary learning (SDL) and SAEs are closely related in that both aim to represent data using sparse latent codes, but they differ fundamentally in how this objective is achieved. SDL is formulated as a strictly linear model in which the data are approximated as a linear combination of dictionary atoms, whereas SAEs replace this sparse-coding step with a nonlinear encoder, typically using ReLU-based activations. In this sense, our theorems can be interpreted as a nonlinear extension of classical dictionary identifiability results. Specifically, from the perspective of identifiability, we can say an SAE is identifiable if the reconstruction of polysemantic features $x_p$ leads to the perfect reconstruction of ground truth feature $x$, i.e., $\tilde{x}_p = x_p \Rightarrow x_m = x$ up to index reordering and zero padding. Then the observed phenomena of feature shrinking and feature vanishing correspond to forms of partial non-identifiability that arise when the sparsity level is low and the nonlinear encoder cannot perfectly disentangle the ground truth features. On the contrary, by Theorems 2 and 3, we prove the identifiability of SAEs under the extreme sparsity condition. Moreover, Theorem 4 shows that the ground truth reconstrction error of SAEs, which can also be viewed as a measure of identifiability, depends on the interference term $(W_p^\top W_p - I)$, which has similar forms to the coherence condition of SDL. In addition, by Theorem 5, we show that introducing additional weights to the interference terms can enhance SAE identifiability.

**Extension to hierarchical feature structures.** From empirical observations of the related works, the general features can be viewed as a combination of the specialized ones. For example, the activation of the general feature "blue" is a combination of activations of specialized features "blue circle", "blue square", and "blue triangle" with positive weights, because ideally, if any of the three specialized feature fires, the general "blue" should fire. Therefore, as a possible extension

of our theoretical framework, we can view the general features $x_h$ as linear combinations of the specialized features $x$ assumed in our submission, i.e. $x_h = W_h x$. Then the polysemantic SAE inputs $x_p = W_p' x_h = W_p' W_h x := W_p x$ still coincide with our theoretical formulations. Note that under this formulation, for any concatenations of general and specialized features $x_{concat}$, there exists a weight matrix $W_p^*$ such that $x_p = W_p^* x_{concat}$. Then according to our theorems, if $x_{concat}$ features are non-overlapping and extremely sparse, they can be fully recovered by SAEs. This can partly explain the feature absorption phenomenon that the general features are absorbed by the overlapping specialized ones, because non-overlapping sparse features are more recoverable.

**Extension to non-linear feature combinations.** We consider the attention-weighted combinations of features. We discuss that as $\mathrm{softmax}(\cdot) \approx \max(\cdot)$, the attention-weighted combinations can be approximately reduced to a linear form about certain features, so the theoretical results in our paper still hold. Specifically, for an attention-weighted combinations of features, we denote the polysemantic feature as

$$\boldsymbol{x}_{p,t} = \sum_{i=1}^{n} a_{t,i} v_i = \sum_{i=1}^{n} \frac{(W_q \boldsymbol{x}_t)^\top (W_k \boldsymbol{x}_i)}{\sum_j (W_q \boldsymbol{x}_t)^\top (W_k \boldsymbol{x}_j)} W_v \boldsymbol{x}_i,$$

where $W_q$, $W_k$, and $W_v$ denote query, key, and value matrices. Note that $\frac{(W_q \boldsymbol{x}_t)^\top (W_k \boldsymbol{x}_i)}{\sum_j (W_q \boldsymbol{x}_t)^\top (W_k \boldsymbol{x}_j)} = \mathrm{softmax}((\boldsymbol{x}_t^\top W_q^\top W_k \boldsymbol{x}_j)_{j \in [n]})_i \approx \mathbf{1}[i = \arg\max_j \boldsymbol{x}_t^\top W_q^\top W_k \boldsymbol{x}_j]$, so we have

$$\boldsymbol{x}_{p,t} \approx \sum_{i=1}^{n} \mathbf{1}[i = \arg\max_j \boldsymbol{x}_t^\top W_q^\top W_k \boldsymbol{x}_j] W_v \boldsymbol{x}_i = W_v \boldsymbol{x}_{\arg\max_j \boldsymbol{x}_t^\top W_q^\top W_k \boldsymbol{x}_j}.$$

Then we have the SAE risk as

$$\mathcal{L}_{\mathrm{SAE}} = \mathbb{E}_{\boldsymbol{x}} \sum_{t=1}^{n} \|\boldsymbol{x}_{p,t} - W_m^\top \sigma(W_m \boldsymbol{x}_{p,t})\|^2$$

$$\approx \mathbb{E}_{\boldsymbol{x}} \sum_{t=1}^{n} \|W_v \boldsymbol{x}_{\arg\max_j \boldsymbol{x}_t^\top W_q^\top W_k \boldsymbol{x}_j} - W_m^\top \sigma(W_m W_v \boldsymbol{x}_{\arg\max_j \boldsymbol{x}_t^\top W_q^\top W_k \boldsymbol{x}_j})\|^2. \tag{40}$$

By similar proofs of Theorems 1-3, if $W_v$ forms certain geometry or has non-positive interferences, then for $t \in [n]$, we have

$$W_m^* = W_v^\top = \arg\min_{W_m} \mathbb{E}_{\boldsymbol{x}} \|\boldsymbol{x}_{p,t} - W_m^\top \sigma(W_m \boldsymbol{x}_{p,t})\|^2,$$

and accordingly $W_m^* = W_v^\top = \arg\min_{W_m} \mathcal{L}_{\mathrm{SAE}}$. With this closed-form solution, we can show that the insights of SAE feature recovery still hold. Moreover, by similar proofs, the insights of Theorems 4-5 also still hold, only replacing $x$ with $\boldsymbol{x}_{\arg\max_j \boldsymbol{x}_t^\top W_q^\top W_k \boldsymbol{x}_j}$ in the mathematic forms.

**Extensions of loss design beyond reweighting.** As Theorem 5 is based on a very general theoretical framework, any design of the $\Gamma$ matrix that shrinks the gap term $W_p^\top \Gamma^\top \Gamma W_p - I_{n \times n}$ could improve the SAE feature recovery. In our submission, we propose the most simple and intuitive reweighting strategy for an example to demonstrate this. Nonetheless, alternative designs are also foreseeable. For example, although $W_p$ is unknown, according to the closed-form solutions derived in Theorems 1-3, we can in turn estimate it using $W_m^\top$. Then we can deliberately estimate a triangular matrix $\Gamma$ to minimize $\|W_p^\top \Gamma^\top \Gamma W_p - I_{n \times n}\|$. A possible risk of this method would be the estimation error of $W_p$ can be large during the early training stages. Another possible alternative direction would be to include additional regularization term $\|\mathrm{trace}(W_p^\top W_p \Gamma^\top \Gamma) - n\|$. Specifically, note that $\mathrm{trace}(W_p^\top \Gamma^\top \Gamma W_p) = \mathrm{trace}(W_p^\top W_p \Gamma^\top \Gamma)$. Then we can design a triangular matrix $\Gamma$ such that $\mathrm{trace}(W_p^\top W_p \Gamma^\top \Gamma) = \mathrm{trace}(I_{n \times n}) = n$, where the cross-feature interference $W_p^\top W_p$ can be directly estimated from data.

## C   EXPERIMENTS DETAILS

### C.1   ADDITIONAL EXPERIMENTS

**Alternative realization for Top-$k$.** We conduct additional validation experiments for Section 5.2.1 with an alternative realization of the Top-$k$ activation. Specifically, the Top-$k$ dimensions are selected

by decomposing the dimensions into $k$ groups and choosing the Top-1 activated dimensions in each group. This realization could largely enhance the computational efficiency by allowing parallel computing. The results are shown in Table 2. The results show that compared with SAE features, WSAE features obtain higher auto-interpretability scores (i.e., stronger monosemanticity) across different situations.

Table 2: Auto-interpretability scores (%) of SAEs trained following different layers (9-11) of Pythia-160M with original SAE and weighted SAE loss.

|                            | 9     | 10    | 11    |
|----------------------------|-------|-------|-------|
| Original SAE               | 69.0  | 61.8  | 56.3  |
| Weighted SAE ($\alpha = 1$) | **71.5** | **63.7** | **59.0** |
| Gains                      | **+2.5** | **+1.9** | **+2.7** |

Table 3: Auto-interpretability scores (%) of Llama-3-8B with original SAE and weighted SAE loss.

| Layer              | 8     | 16    | 24    | 32    |
|--------------------|-------|-------|-------|-------|
| SAE                | 62.0  | 67.0  | 66.7  | 61.4  |
| WSAE ($\alpha = 1$) | **66.5** | **80.2** | **70.5** | **63.6** |
| Gain               | **+4.5** | **+2.8** | **+3.8** | **+2.2** |

**Evaluation on Llama.** We conduct additional experiments on Llama-3-8B and present the last-layer results in Table 3. The results show that compared with SAE features, WSAEs show a consistent improvement in interpretability compared with the original SAE across different layers, which demonstrates the effectiveness of our proposed WSAE. This result coincides with our result on the smaller Pythia-160M.

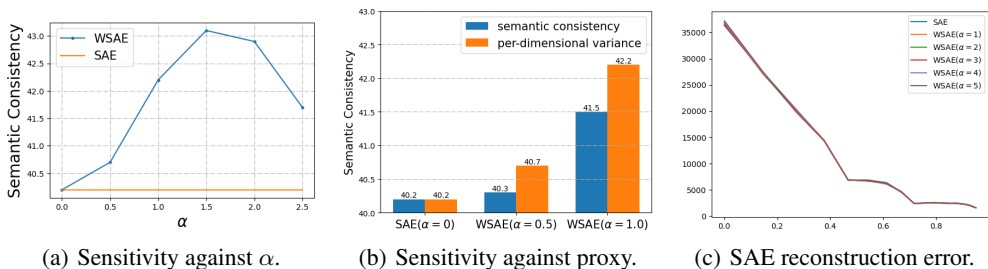

(a) Sensitivity against $\alpha$.  (b) Sensitivity against proxy.  (c) SAE reconstruction error.

Figure 5: (a) Semantic consistency of WSAEs under different $\alpha$. (b) Semantic consistency of SAE and WSAEs with different monosemanticity proxies, including semantic consistency and per-dimensional variance. (c) SAE reconstruction error of WSAEs under different $\alpha$.

**Sensitivity analysis regarding $\alpha$.** We run experiments with $alpha \in \{0, 0.5, 1.0, 1.5, 2.0, 2.5\}$ and report the semantic consistency. The other settings are kept exactly the same with that of Figure 4 in our submission. The results are shown in Figure 5(a). We show that in wide selection range of $\alpha$, our WSAE show a consistent advantage over SAE ($\alpha = 0$) with respect to interpretability, which demonstrates the robustness of WSAE against $\alpha$.

**Sensitivity analysis regarding monosemanticity proxy.** We conduct an empirical comparison of different proxies ($s_i$) (variance vs. semantic metrics) to test transferability. Specifically, as an extension of Figure 3, we replace the semantic consistency with per-dimensional variance. The results are reported in Figure 5(b). We show that WSAEs with the two proxies both show a consistent advantage over SAE ($\alpha = 0$) with respect to interpretability, and the variation trend against $\alpha$ is also similar. This demonstrates the robustness of WSAE against proxy selection of monosemanticity.

**Influence of reweighting on the Pareto frontier.** In Figure 5(c), we show the SAE reconstruction error of $x_p$ of WSAEs under different $\alpha$'s on the synthetic dataset, where the $x$-axis is the sparsity level of the ground truth features. We keep the other settings the same as Figure 3. We show that the SAE reconstruction errors of WSAEs are approximately the same as those of the original SAE, and

the results remain robust across a relatively wide range of $\alpha$ selections. This indicates that WSAEs still roughly lie on the Pareto frontier.

## C.2 COMPUTE RESOURCES

For the experiments on toy models in Section 4, we conduct experiments on an NVIDIA 3090 GPU with 24GB of memory. The training and evaluation of a toy model takes around 30 minutes. For the experiments on language models in Section 5.1, we conduct experiments on an NVIDIA A100 GPU with 40GB of memory. The training of a sparse autoencoder takes around 24 hours, and the evaluation needs 10 minutes. While for experiments on vision models in Section 5.2, we conduct experiments on two NVIDIA 3090 GPUs with 24GB of memory. The training of a sparse autoencoder takes around 12 hours, and the evaluation needs 5 minutes.

## C.3 AVERAGE ACTIVATED FEATURES

For a sample $x \in \mathbb{R}^n$, we perform $n$ encodings, each time isolating a single input feature. Specifically, for the $i$-th encoding, we construct $x^i = (0, \cdots, x_i, \cdots, 0)$, where only the $i$-th feature is retained. This yields $n$ SAE latent representations $\{h(x^i)\}_{i=1}^n$, each corresponding to the activation induced by a single input feature. We then calculate the total activation values across different samples in each SAE latent dimension. For example, for $j$-th dimension in SAE latents, we define $M_j^i = \sum_x h_j(x^i)$ representing the cumulative activation from the $i$-th input feature. If the value $M_j^i$ exceeds a threshold, we consider the $j$-th SAE latent to be activated by the $i$-th input. Finally, we compute the average number of input features that activate each SAE latent dimension. We note that if the SAE successfully recovers monosemantic features, each latent should be activated by only a small number of input features.

## C.4 AN AUTOINTERPRETATION PROTOCOL

The autointerpretability process consists of five steps and yields both an interpretation and an autointerpretability score:

1. On each of the first 50,000 lines of OpenWebText, take a 64-token sentence-fragment, and measure the feature's activation on each token of this fragment. Feature activations are rescaled to integer values between 0 and 10.

2. Take the 20 fragments with the top activation scores and pass 5 of these to GPT-4, along with the rescaled per-token activations. Instruct GPT-4 to suggest an explanation for when the feature (or neuron) fires, resulting in an interpretation.

3. Use GPT-3.56 to simulate the feature across another 5 highly activating fragments and 5 randomly selected fragments (with non-zero variation) by asking it to provide the per-token activations. While the process described in Bills et al. (2023) uses GPT-4 for the simulation step, we use GPT-3.5. This is because the simulation protocol requires the model's logprobs for scoring, and OpenAI's public API for GPT-3.5 (but not GPT-4) supports returning logprobs.

4. Compute the correlation of the simulated activations and the actual activations. This correlation is the autointerpretability score of the feature. The texts chosen for scoring a feature can be random text fragments, fragments chosen for containing a particularly high activation of that feature, or an even mixture of the two. We use a mixture of the two unless otherwise noted, also called 'top-random' scoring.

5. If, amongst the 50,000 fragments, there are fewer than 20 which contain non-zero variation in activation, then the feature is skipped entirely.

