# OpenReview forum: "On the Limits of Sparse Autoencoders: A Theoretical Framework and Reweighted Remedy"
_ICLR.cc/2026/Conference — ICLR 2026 Poster_

### Official Review · Reviewer_hKP4 · 2025-10-31

**Soundness:** 4
**Presentation:** 3
**Contribution:** 4
**Rating:** 8
**Confidence:** 3

**Summary:**

This paper presents the first theoretical framework analyzing the fundamental limits of Sparse Autoencoders (SAEs) for recovering monosemantic features from superposed polysemantic representations. The authors derive closed-form solutions showing that, under general conditions, SAEs cannot perfectly recover ground-truth monosemantic features due to feature shrinking and feature vanishing effects. They identify that full recovery is theoretically guaranteed only under extreme sparsity of the true features. To address this limitation, the paper introduces a Weighted Sparse Autoencoder (WSAE) that reweights the reconstruction loss according to the degree of polysemanticity of each input dimension. Theoretical analysis demonstrates that this weighting narrows the gap between the SAE loss and the ideal ground-truth reconstruction loss, and experiments on both synthetic data and pretrained models confirm that WSAE improves feature monosemanticity and interpretability without sacrificing reconstruction fidelity.

**Strengths:**

The paper’s originality lies in providing a formal, mathematical explanation for why SAEs sometimes fail to identify interpretable features, moving beyond the purely empirical understanding that dominates current mechanistic interpretability work. The authors establish clear analytical results—closed-form solutions, necessary and sufficient conditions, and a uniqueness theorem—that rigorously connect feature sparsity with successful monosemantic recovery. This theoretical grounding fills a long-standing gap in the field, where SAE behavior had been empirically impressive but conceptually opaque. The proposed reweighting principle is both simple and theoretically motivated, making it an elegant bridge between formal analysis and practical implementation.

In terms of quality and clarity, the paper is meticulously written, with precise mathematical derivations and well-structured proofs. The relationship between the SAE loss and the ideal ground-truth loss is clearly articulated and forms a compelling narrative that unifies the theoretical and empirical sections. The significance of the contribution is substantial: by formalizing the limitations of current interpretability techniques, the paper reframes the field’s expectations of what SAEs can and cannot achieve, and introduces a principled path forward through reweighted optimization. The experiments, while modest in scale, convincingly validate the theoretical predictions and demonstrate that the framework generalizes across modalities.

**Weaknesses:**

The main limitation is the scope of empirical validation. The experiments are primarily performed on small or medium-scale models (Pythia-160M, ResNet-18) and under controlled settings. While these choices are appropriate for validating the theory, it remains unclear how well the findings extend to large modern LLMs or to deeper, multi-layer SAE architectures that are increasingly used in practice. Furthermore, the superposition assumption in the theoretical model treats representations as linear mixtures, which simplifies the nonlinear interactions and attention-based dynamics found in real networks. As a result, the framework captures the core geometry of superposition but may not fully describe the behavior of realistic LLM feature spaces.

**Questions:**

1. How does the theoretical framework extend to non-linear or multi-layer autoencoders, especially when feature mixing occurs across layers? Could the authors discuss whether similar recovery limits hold in deeper settings?

2. The analysis assumes linear superposition of features. Have the authors explored how deviations from linearity—such as attention-weighted combinations or nonlinear feature interactions—affect the recovery bounds?

---

> ### Author Response · Authors · 2025-11-21
>
> We express our sincere gratitude to Reviewer hKP4 for appreciating the originality, clarity, and significance of our theoretical framework. We address your concerns below.
>
> ---
>
> **W1.** The main limitation is the scope of empirical validation. The experiments are primarily performed on small or medium-scale models (Pythia-160M, ResNet-18) and under controlled settings. While these choices are appropriate for validating the theory, it remains unclear how well the findings extend to large modern LLMs or to deeper, multi-layer SAE architectures that are increasingly used in practice.
>
> **A.** Following your suggestions, **we run additional experiments on the larger Llama3-8b model**, and report the layer-wise comparisons of the auto-interpretability scores between SAE and WSAE, with results reported in the following table. We see that WSAE shows a consistent improvement in interpretability compared with the original SAE across different layers, which demonstrates the effectiveness of our proposed WSAE. This result coincides with our conclusions in Section 5.2.1.
>
> | Layer | 8 | 16 | 24 | 32 |
> |--------------|--------------|-----------|-----------|-----------|
> | SAE  | 62.0 | 67.0 | 66.7 | 61.4 |
> | WSAE | **66.5** | **80.2** | **70.5** | **63.6** |
> | Gain | **+4.5** | **+2.8** | **+3.8** | **+2.2** |
>
> ---
>
> **W2.** The superposition assumption in the theoretical model treats representations as linear mixtures, which simplifies the nonlinear interactions and attention-based dynamics found in real networks. As a result, the framework captures the core geometry of superposition but may not fully describe the behavior of realistic LLM feature spaces.
>
> **A.** Following your suggestions, we additionally discuss the multi-layer feature structure and the non-linear mixtures of features in the answers to **Q1** and **Q2**, respectively.
>
> ---
>
> **Q1.** How does the theoretical framework extend to non-linear or multi-layer autoencoders, especially when feature mixing occurs across layers? Could the authors discuss whether similar recovery limits hold in deeper settings?
>
> **A.** We discuss the results of our theoretical framework under **multi-layer linear autoencoders**, and we leave the generalizations of the non-linear cases to the answers to **Q2**. For multi-layer autoencoders, when feature mixing occurs across layers, the early-layer features tend to be specialized features (e.g. "Lily", "Sarah", "Mary", etc.), whereas the latter-layer features, as combinations of the specialized features, tend to be general features (e.g. "female names"). That is, in this case, **we naturally extend our theoretical framework to mixing features with hierarchical structures**. Specifically, take a linear 3-layer case as an example. We denote $x$ as the early-layer ground-truth (specialized) features, and $x_h$ as the latter-layer (general) features. Then we have $x_h=W_h x$. And we have the final-layer polysemantic features $x_p = W_p'x_h = W_p' W_h x := W_p x$ serving as the input to SAEs. Note that under this formulation, for any concatenations of mixing-layer features $x\_{concat}$ composed of elements in both $x$ and $x_h$, there exists a weight matrix $W_p^*$ such that $x_p = W_p^{\*} x\_{concat}$. Then according to our theory, **if $x\_{concat}$ features are non-overlapping and extremely sparse, they can still be fully recovered by SAEs.**
>
> In our revised version, we have also added a paragraph in a newly added discussion section (Appendix B) to discuss these potential extensions.

---

> ### Author Response · Authors · 2025-11-21
>
> **Q2.** The analysis assumes linear superposition of features. Have the authors explored how deviations from linearity—such as attention-weighted combinations or nonlinear feature interactions—affect the recovery bounds?
>
> **A.** **We take the attention-weighted combinations as an example of non-linear generalizations of our theoretical framework**. We discuss that as $\mathrm{softmax}(\cdot)\approx \mathrm{max}(\cdot)$, the attention-weighted combinations can be approximately reduced to a linear form about certain features, so **the theoretical results in our paper still hold**. In our revised version, we have also included this potential extension in a newly added discussion section (Appendix B).
>
> Specifically, for an attention-weighted combination of features, we denote the polysemantic feature as
> $$
> \boldsymbol{x}\_{p,t} = \sum\_{i=1}^n a\_{t,i} v_i = \sum\_{i=1}^n \frac{(W_q\boldsymbol{x}\_t)^\top (W_k \boldsymbol{x}\_i)}{\sum_j (W_q\boldsymbol{x}\_t)^\top (W_k \boldsymbol{x}\_j)}W_v \boldsymbol{x}\_i,
> $$
> where $W_q$, $W_k$, and $W_v$ denote query, key, and value matrices.
>
> Note that $\frac{(W_q\boldsymbol{x}\_t)^{\top} (W_k \boldsymbol{x}\_i)}{\sum\_j (W_q\boldsymbol{x}\_t)^{\top} (W_k \boldsymbol{x}\_j)}=\mathrm{softmax}((\boldsymbol{x}\_t^{\top} W_q^{\top} W_k \boldsymbol{x}\_j)\_{j\in [n]})\_i\approx \boldsymbol{1}[i=\arg\max\_j \boldsymbol{x}\_t^{\top} W_q^{\top} W_k \boldsymbol{x}\_j]$, so we have
> $$
> \boldsymbol{x}\_{p,t} \approx \sum\_{i=1}^n \boldsymbol{1}[i=\arg\max_j \boldsymbol{x}\_t^{\top} W_q^{\top} W_k \boldsymbol{x}\_j] W_v \boldsymbol{x}\_i = W_v \boldsymbol{x}\_{\arg\max_j \boldsymbol{x}\_t^{\top} W_q^\{top} W_k \boldsymbol{x}\_j}.
> $$
> Then we have the SAE risk as
> $$
> \mathcal{L}\_{\mathrm{SAE}} = \mathbb{E}\_{\boldsymbol{x}} \sum\_{t=1}^n \\|\boldsymbol{x}\_{p,t} - W_m^{\top} \sigma(W_m\boldsymbol{x}\_{p,t})\\|^2 \approx \mathbb{E}\_{\boldsymbol{x}} \sum\_{t=1}^n \\|W_v \boldsymbol{x}\_{\arg\max_j \boldsymbol{x}\_t^\top W_q^{\top} W_k \boldsymbol{x}\_j} - W_m^\top \sigma(W_m W_v \boldsymbol{x}\_{\arg\max_j \boldsymbol{x}\_t^\top W_q^\top W_k \boldsymbol{x}\_j})\\|^2.
> $$
> By similar proofs of Theorems 1-3, if $W_v$ forms certain geometry or has non-positive interferences, then for $t \in [n]$, we have
> $$
> W_m^* = W_v^{\top} = \arg\min\_{W_m} \mathbb{E}\_{\boldsymbol{x}} \\|\boldsymbol{x}\_{p,t} - W_m^{\top} \sigma(W_m\boldsymbol{x}\_{p,t})\\|^2,
> $$
> and accordingly $W_m^* = W_v^\top = \arg\min_{W_m} \mathcal{L}_{\mathrm{SAE}}$.
>
> With this closed-form solution, we can show that the insights of SAE feature recovery still hold. Moreover, by similar proofs, the insights of Theorems 4-5 also still hold, only replacing $x$ with $\boldsymbol{x}_{\arg\max_j \boldsymbol{x}_t^\top W_q^\top W_k \boldsymbol{x}_j}$ in the mathematic forms.
>
> ---
>
> Thank you for your insightful and constructive comments. Hope our explanations and additional experiments can address your concerns.

---

> ### Author Response · Authors · 2025-11-27
> **Could you please provide feedback on our rebuttal?**
>
> Dear Reviewer,
>
> We have done our best to address all your comments in the rebuttal. As the discussion period is ending in less than a week, we would greatly appreciate it if you could take a moment to review our responses and let us know whether our reply sufficiently resolved your concerns.
>
> Thank you very much for your time and consideration.
>
> Best regards,
>
> Authors

---

### Official Review · Reviewer_TYYe · 2025-11-01

**Soundness:** 3
**Presentation:** 3
**Contribution:** 3
**Rating:** 6
**Confidence:** 3

**Summary:**

This paper provides a rigorous theoretical analysis of the **feature recovery limits of Sparse Autoencoders (SAEs)**, a method widely used for interpreting polysemantic representations in large models.
Under the **superposition hypothesis**, the authors derive a closed-form optimal solution for SAEs and prove that:
- In general, SAEs cannot perfectly recover the true monosemantic features due to *feature shrinking* and *feature vanishing* effects.
- Full recovery is guaranteed only under **extremely sparse ground-truth features** (sparsity factor \(S \to 1\)), i.e., nearly 1-sparse activations.
- To mitigate this, the paper proposes a **Weighted Sparse Autoencoder (WSAE)**, where per-dimension weights reduce the reconstruction gap between the observed polysemantic and unobserved monosemantic features.
Theoretical results (Theorem 4–5) show that proper weighting minimizes this gap, and experiments on synthetic data and pretrained models (Pythia-160M, ResNet-18) demonstrate improved interpretability metrics.

**Strengths:**

- **Mathematically grounded analysis.**
  The derivation of a closed-form SAE solution and identification of feature shrinking/vanishing phenomena clarify long-standing empirical observations in mechanistic interpretability.
  While related to classical sparse coding theory, the explicit analytical form for the ReLU-based SAE and the formal characterization of these degradation modes constitute a novel operational understanding that was not previously formalized.

- **Bridging theory and practice.**
  The framework connects abstract sparse recovery theory to interpretability practice in LLMs, offering insight into why SAEs sometimes fail to yield cleanly separable features.

- **Theoretically motivated improvement.**
  The proposed WSAE provides a principled approach to reweight reconstruction according to estimated polysemanticity, improving monosemanticity without large losses in reconstruction quality (as evidenced in Fig.3(c)).

- **Transparent proofs and clear assumptions.**
  All mathematical steps, assumptions (superposition, sparsity distribution), and limitations are clearly stated.

**Weaknesses:**

### 1. Relationship to dictionary learning and identifiability
The theoretical results closely parallel classical **identifiability conditions in sparse dictionary learning**, where full recovery requires incoherence or extreme sparsity of the underlying basis.
While this correspondence is intuitive, it is not explicitly discussed in the paper.
Clarifying this relationship would enhance the theoretical positioning of the work.

In particular:
- Theorem 1–3 can be viewed as a **nonlinear (ReLU-based) extension of dictionary identifiability** results, where interference \(W_p^\top W_p - I\) plays the role of coherence.
- The observed feature shrinking/vanishing could be interpreted as manifestations of partial non-identifiability under finite sparsity.

Explicitly situating the paper in relation to established sparse coding theory (e.g., spark condition, mutual coherence) would help readers understand which parts of the contribution are new—e.g., the inclusion of ReLU nonlinearities and overcomplete encoders—and which are theoretical refinements of existing principles.

---

### 2. Implications for interpretability and practical impact
The paper’s findings—that SAEs cannot fully disentangle superposed representations except in extreme sparsity—have significant consequences for current interpretability research, though this point could be emphasized more strongly.

In particular:
- Many interpretability studies assume that increasing sparsity or width of SAEs improves feature separation indefinitely.
  This work shows that such improvement **plateaus due to intrinsic representational interference**, meaning full disentanglement is mathematically impossible under realistic sparsity.
- Consequently, **SAE-based interpretability should be regarded as an approximation tool**, not as a faithful feature recovery mechanism.
  This reframes SAE-derived neurons as *approximate projections of overlapping features*, rather than direct encodings of ground-truth concepts.

This reinterpretation could reshape how SAE-based analyses are used in mechanistic interpretability: rather than aiming for perfectly monosemantic neurons, practitioners might instead quantify or visualize residual interference between features.

A related empirical suggestion would be to measure the effective sparsity \(S\) of real LLM activations, to contextualize how close such models operate to the theoretical extreme-sparsity regime.

---

### 3. Sensitivity and robustness of the reweighting scheme
The WSAE introduces weights \(\gamma_i = s_i^{\alpha}\), with \(s_i\) estimated from variance or semantic consistency and \(\alpha\) controlling emphasis on monosemantic dimensions.
The authors note (p.6) that results are “relatively robust” to α and show examples for α = 0.5 and 1.0 (Fig. 4), which supports this claim.
Nonetheless, a small sensitivity analysis or ablation could strengthen the argument.

Suggestions:
- A sweep of α (e.g., {0, 0.5, 1, 2}) to illustrate stability trends.
- A comparison of different proxies \(s_i\) (variance vs. semantic metrics) to test transferability.
- Discussion of reconstruction trade-offs: Fig.3(c) indicates that \(x_p\) reconstruction is maintained (i.e., no major Pareto penalty), but confirming this across datasets would reinforce generality.

These additions would verify that the reported improvements are not dataset-specific and that α tuning is unnecessary in practice.

---

### 4. Reproducibility and release
Appendix B.2 clearly reports compute resources and training time (e.g., 24 h on A100 for language models).
It would still be helpful to confirm whether **code and theoretical implementations** will be released upon publication, enabling the community to replicate and extend the analysis.

---

### 5. Potential extensions
The theoretical framework appears extensible beyond reweighting—for instance, to alternative loss formulations or nonlinear encoder architectures that directly address the feature interference term \(W_p^\top W_p - I\).
A short remark in the final version about such future directions would underscore the broader applicability of this analysis.

**Questions:**

1. How do Theorem 1–3 connect formally to existing identifiability results in sparse dictionary learning (e.g., spark or coherence conditions)?
2. How sparse are real LLM activations compared to the “extreme sparsity” regime analyzed here?
3. Could you include a small α-sweep or sensitivity plot to verify robustness?
4. Does reweighting ever degrade reconstruction accuracy, or is the Pareto frontier generally preserved (as suggested by Fig. 3(c))?
5. Will the theoretical framework and WSAE implementation be publicly released?
6. Do you foresee extensions of this theoretical setup beyond reweighting—for example, alternative matrix designs that directly minimize cross-feature interference?

---

> ### Author Response · Authors · 2025-11-21
>
> We express our sincere gratitude to Reviewer TYYe for appreciating our rigorous theoretical analysis and our theoretically motivated improvement. We address your concerns below.
>
> ---
>
> **W1.** Relationship to dictionary learning and identifiability.
>
> **A.** Following your suggestions, we discuss the relationship of our paper to sparse dictionary learning and identifiability. Sparse dictionary learning (SDL) and SAEs are closely related in that **both aim to represent data using sparse latent codes, but they differ fundamentally in how this objective is achieved**. SDL is formulated as a strictly linear model in which the data are approximated as a linear combination of dictionary atoms, whereas SAEs replace this sparse-coding step with a nonlinear encoder, typically using ReLU-based activations. In this sense, **our theorems can be interpreted as a nonlinear extension of classical dictionary identifiability results**. Specifically, from the perspective of identifiability, we can say an SAE is identifiable if the reconstruction of polysemantic features $x_p$ leads to the perfect reconstruction of ground truth feature $x$, i.e., $\tilde{x}_p=x_p\ \Rightarrow\ x_m=x$ up to index reordering and zero padding. Then **the observed phenomena of feature shrinking and feature vanishing correspond to forms of partial non-identifiability** that arise when the sparsity level is low and the nonlinear encoder cannot perfectly disentangle the ground truth features. On the contrary, by Theorems 2-3, we prove the identifiability of SAEs under the extreme sparsity condition. Moreover, our Theorem 4 shows that the ground truth reconstruction error of SAEs, which **can also be viewed as a measure of identifiability**, depends on the interference term ($W_p^\top W_p-I$), which **has a similar form to the coherence condition of SDL**. In addition, by Theorem 5, we show that introducing additional weights to the interference terms can enhance SAE identifiability.
>
> In our revised version, we have also added a paragraph in a newly added discussion section (Appendix B) to make this connection explicit.
>
> ---
>
> **W2.** Implications for interpretability and practical impact.
>
> **A.** Thank you for the valuable suggestions. In our revised version, we have added the discussions about the significance of our work in the conclusion section.
>
> Specifically, we write "While many interpretability studies assume that increasing sparsity or width of SAEs improves feature separation indefinitely, this work shows that such improvement plateaus due to intrinsic representational interference, meaning full disentanglement is mathematically impossible under realistic sparsity. Consequently, SAE-based interpretability should be regarded as an approximation tool, not as a faithful feature recovery mechanism. This reframes SAE-derived neurons as approximate projections of overlapping features, rather than direct encodings of ground-truth concepts."
>
> For the empirical suggestion regarding sparsity, please refer to our reply to **Q2**.

---

> ### Author Response · Authors · 2025-11-21
>
> **W3.** Sensitivity and robustness of the reweighting scheme.
>
> **A.** Following your suggestion **regarding stability trends**, we run experiments with $\alpha \in \\{0, 0.5, 1.0, 1.5, 2.0, 2.5\\}$ and report the semantic consistency. The other settings are kept exactly the same as those of Figure 4 in our submission. The results are shown in the following table and in the additional Figure 5(a) in Section B.1. We show that in a wide selection range of $\alpha$, our WSAE shows a consistent advantage over SAE ($\alpha=0$) with respect to interpretability, which demonstrates the robustness of WSAE against $\alpha$.
>
> | $\alpha$ | 0 | 0.5 | 1.0 | 1.5 | 2.0 | 2.5 |
> |--------------|--------------|-----------|-----------|-----------|-----------|-----------|
> | semantic consistency  | 40.2 | 40.7 | 42.2 | 43.1 | 42.9 | 41.7 |
>
> Following your suggestion **regarding different proxies**, we conduct an empirical comparison of different proxies (s_i) (variance vs. semantic metrics) to test transferability. Specifically, as another extension of Figure 4, we replace the semantic consistency with per-dimensional variance. The results are reported in the following table and the additional Figure 5(b) in Section B.1. We show that WSAEs with the two proxies both show a consistent advantage over SAE ($\alpha=0$) with respect to interpretability, and the variation trend against $\alpha$ is also similar. This demonstrates the robustness of WSAE against proxy selection of monosemanticity.
>
> | $\alpha$ | 0 | 0.5 | 1.0 |
> |--------------|--------------|-----------|-----------|
> | semantic consistency  | 40.2 | 40.7 | 42.2 |
> | per-dimensional variance | 40.2 | 40.3 | 41.5 |
>
> Following your suggestion **regarding reconstruction trade-offs**, we conduct additional experiments calculating the SAE reconstruction error of WSAEs with different values of $\alpha$, and present the results in this anonymous link https://imgur.com/a/dlyR3o3 and also Figure 5 of our revised manuscript. We show that the SAE reconstruction errors of WSAEs are approximately the same as those of the original SAE, and the results remain robust across a relatively wide range of $\alpha$ selections. This indicates that WSAEs still roughly lie on the Pareto frontier.
>
> ---
>
> **W4.** Reproducibility and release.
>
> **A.** As mentioned in the reproducibility statement, we will release our code ASAP upon publication. Also, all details of our theoretical framework are presented in our submission with all proofs in Section A to guarantee reproducibility.
>
> ---
>
> **W5.** Potential extensions.
>
> **A.** As Theorem 5 is based on a very general theoretical framework, any design of the $\Gamma$ matrix that shrinks the gap term $W_p^\top\Gamma^\top \Gamma W_p - I_{n\times n}$ could improve the SAE feature recovery. In our submission, we propose the simplest and intuitive reweighting strategy for an example to demonstrate this. Nonetheless, alternative designs are also foreseeable. For example, although $W_p$ is unknown, according to the closed-form solutions derived in Theorems 1-3, we can in turn estimate it using $W_m^\top$. Then **we can deliberately estimate a triangular matrix $\Gamma$ to minimize $\\|W_p^\top\Gamma^\top \Gamma W_p - I_{n\times n}\\|$**. A possible risk of this method would be that the estimation error of $W_p$ can be large during the early training stages. **Another possible alternative direction would be to include an additional regularization term $\\|\mathrm{trace}(W_p^\top W_p \Gamma^\top \Gamma)-n\\|$**. Specifically, note that $\mathrm{trace}(W_p^\top \Gamma^\top \Gamma W_p) = \mathrm{trace}(W_p^\top W_p \Gamma^\top \Gamma)$. Then we can design a triangular matrix $\Gamma$ such that $\mathrm{trace}(W_p^\top W_p \Gamma^\top \Gamma)=\mathrm{trace}(I_{n\times n})=n$, where the cross-feature interference $W_p^\top W_p$ can be directly estimated from data.
>
> In our revised version, we have also added a paragraph in a newly added discussion section (Appendix B) to discuss these potential extensions.

---

> ### Author Response · Authors · 2025-11-21
>
> **Q1.** How do Theorem 1–3 connect formally to existing identifiability results in sparse dictionary learning (e.g., spark or coherence conditions)?
>
> **A.** We have discussed the relationship of our paper to sparse dictionary learning and identifiability. For more details, please refer to our reply to **W1** and Appendix B in our revised manuscript.
>
> ---
>
> **Q2.** How sparse are real LLM activations compared to the “extreme sparsity” regime analyzed here?
>
> **A.** We respectfully note that there seems to be a misunderstanding about the “extreme sparsity” condition. **The LLM activations (inputs of SAEs) are not assumed to be sparse, whereas the “extreme sparsity” of the underlying ground truth features, which are theoretically assumed but inherently agnostic, is required for SAE full recovery**. As the ground truth features of real LLMs are inherently unknown, in our submission, we validate SAE feature recovery w.r.t. sparsity via synthetic experiments. Specifically, in Figure 3, we validate that the extremely sparse latents ($S \to 1$) are fully recoverable by SAEs, whereas the less sparse ones would have larger ground truth reconstruction error.
>
> ---
>
> **Q3.** Could you include a small α-sweep or sensitivity plot to verify robustness?
>
> **A.** Yes, we have included sensitivity plots regarding both $\alpha$-sweep and monosemanticity proxies to verify robustness. Please refer to our reply to **W3** for more details.
>
> ---
>
> **Q4.** Does reweighting ever degrade reconstruction accuracy, or is the Pareto frontier generally preserved (as suggested by Fig. 3(c))?
>
> **A.** The Pareto frontier is generally preserved. We conduct additional experiments calculating the SAE reconstruction error of WSAEs with different values of $\alpha$. Please refer to our reply to **W3** for more details.
>
> ---
>
> **Q5.** Will the theoretical framework and WSAE implementation be publicly released?
>
> **A5.** Yes, we will release our code ASAP upon publication.
>
> ---
>
> **Q6.** Do you foresee extensions of this theoretical setup beyond reweighting—for example, alternative matrix designs that directly minimize cross-feature interference?
>
> **A.** Yes, we have discussed potential alternative designs, including triangular matrix estimation and additional regularization. Please refer to our reply to **W5** for more details.
>
> ---
>
> Thank you for your insightful and constructive comments. Hope our explanations and additional experiments can address your concerns.

---

> ### Author Response · Authors · 2025-11-27
> **Could you please provide feedback on our rebuttal?**
>
> Dear Reviewer,
>
> We have done our best to address all your comments in the rebuttal. As the discussion period is ending in less than a week, we would greatly appreciate it if you could take a moment to review our responses and let us know whether our reply sufficiently resolved your concerns.
>
> Thank you very much for your time and consideration.
>
> Best regards,
>
> Authors

---

### Official Review · Reviewer_DB94 · 2025-11-01

**Soundness:** 2
**Presentation:** 2
**Contribution:** 2
**Rating:** 4
**Confidence:** 3

**Summary:**

This paper provides a theoretical analysis of SAEs which motivates an alternative SAE design proposal (WSAE). They claim WSAE improves feature mono-semanticity and interpretability.

**Strengths:**

- The paper provides a detailed theoretical analysis.
- The paper proposes a new re-weighting strategy that may reduce polysemanticity.
- the paper tests their new strategy on language models.

**Weaknesses:**

- The paper assumes an incorrect model of the underlying data distribution which is no longer considered valid by many researchers in the field. There is not one true set of non-overlapping features, but rather many kinds of features which overlap with one another. Additionally, many features, such as parts of speech are dense. While much work on SAEs has assumed sparse features in a non-overlapping basis - this was more reasonable to do a few years ago before we'd seen so much object level data. For example, see "Sparse Autoencoders Do Not Find Canonical Units of Analysis" or "A is for Absorption: Studying Feature Splitting and Absorption in Sparse Autoencoders" and indeed some SAE architectures have been proposed with this in mind "Learning Multi-Level Features with Matryoshka Sparse Autoencoders".

- Empirical results on real data are filtered through the lens of an auto-interpretability scoring method which doesn't reflect the broader quality of an SAE.

Figure 4 deliberately scales the x-axis to make minor differences in the semantic consistency metric look very large. It's unclear if this is meaningful or not.

**Questions:**

- I'd like to see theoretical analysis which assumes more complex underlying data distributions. Would this change the resulting conclusions or insights?
- Use of established, if imperfect benchmarks like SAE bench could be used to provide a more comprehensive endorsement of the wSAE approach. For example, do wSAEs demonstrate less feature absorption?

---

> ### Author Response · Authors · 2025-11-21
>
> We express our sincere gratitude to Reviewer DB94 for appreciating our theoretical significance and our new method. We address your concerns below.
>
> ---
>
> **W1.** The paper assumes an incorrect model of the underlying data distribution which is no longer considered valid by many researchers in the field. There is not one true set of non-overlapping features, but rather many kinds of features which overlap with one another. Additionally, many features, such as parts of speech are dense. While much work on SAEs has assumed sparse features in a non-overlapping basis - this was more reasonable to do a few years ago before we'd seen so much object level data.
>
> **A.** **We respectfully note that there might be some misunderstandings about the scope of our paper.** We understand your concern on our mathematical modeling of non-overlapping features, as the listed papers show that general features and specialized features could overlap (e.g. the general feature "female names" and the specialized feature "Lily" overlap). However, in our paper, we aim to study whether a large enough SAE can recover the ground truth features, so **in this large-SAE case, we only need to focus on the recovery of the specialized features, which are inherently non-overlapping** (e.g. specialized features "Lily" and "Sarah" do not overlap). Therefore, in this case, our non-overlapping feature assumption is correct and "captures the core geometry of superposition" as also recognized by Reviewer hKP4. Moreover, **our theoretical framework can be easily extended to overlapping features with only slight modifications to the feature generation procedure**, and can partly explain the feature absorption phenomenon observed in the listed papers. Furthermore, **our framework is also compatible with dense features**, the existence of which exactly motivates us to propose WSAE. In the following, we address your concern in greater detail.
>
> First of all, **the listed papers show that the SAE learned features can have hierarchical structures**, where the general features (parent features) could overlap with the specialized features (child features), or two general features could overlap. For example, the general feature "blue" overlaps with the specialized feature "blue circle" because when "blue circle" activates, "blue" should also activate. Also, the general feature "blue" could also overlap with the general feature "circle" because when "blue circle" activates, both "blue" and "circle" should activate. However, **when the hidden dimension of an SAE is large enough, the general features split into specialized (non-overlapping) features** (e.g. the general feature "female names" splits into specialized features "Lily", "Sarah", "Mary", etc.). In our theorems, since we assume that $n_m \geq n$, we only need to model the specialized (non-overlapping) features and see if they can be fully recovered by SAEs.
>
> Secondly, **our theoretical framework can be easily extended to overlapping features with hierarchical structures.** Specifically, from empirical observations of the related works, the general features can be viewed as a combination of the specialized ones. (Although Leask et. al. 2025 argued that large-SAE features (specialized features) are compositions of small-SAE features (general features), it is mathematically the other way around. E.g. the activation of the general feature "blue" is a combination of activations of specialized features "blue circle", "blue square", and "blue triangle" with positive weights, because ideally, if any of the three specialized features fires, the general "blue" should fire.) Therefore, as a possible extension of our theoretical framework, we can view the general features $x_h$ as linear combinations of the specialized features $x$ assumed in our submission, i.e. $x_h = W_hx$. Then the polysemantic SAE inputs $x_p = W_p'x_h = W_p' W_h x := W_p x$ still coincide with our theoretical formulations. Note that under this formulation, for any concatenations of general and specialized features $x\_{concat}$, there exists a weight matrix $W_p^\*$ such that $x_p = W_p^* x\_{concat}$. Then according to our theorems, if $x\_{concat}$ features are non-overlapping and extremely sparse, they can be fully recovered by SAEs. **This can partly explain the feature absorption phenomenon that the general features are absorbed by the overlapping specialized ones, because non-overlapping sparse features are more recoverable.**
>
> Last but not least, **our framework is also compatible with dense features, and the existence of dense features is exactly our motivation to propose WSAE**. Specifically, in Sections 4.1 and 4.2, we theoretically demonstrate that SAEs fail to recover the ground truth monosemantic features unless the underlying features are extremely sparse. Then in Section 4.3, we propose the reweighting strategy (WSAE) to enhance the effectiveness of SAEs when the underlying features are less sparse.

---

> ### Author Response · Authors · 2025-11-21
>
> **W2.** Empirical results on real data are filtered through the lens of an auto-interpretability scoring method which doesn't reflect the broader quality of an SAE.
>
> **A.** Following your advice (in **Q2**), we conduct additional experiments about feature absorption of SAEs and WSAEs $(\alpha=1)$ trained on Pythia-160M following SAEbench. The layer-wise absorption score (\%) is reported in the following table, where the lower absorption score indicates better feature absorption. From the experimental results, we show that **the absorption score of WSAEs is better than that of SAEs in most cases**. We conjecture that this is because WSAEs give higher weights to the relatively monosemantic features in $x_p$. As the general features are more likely to be well preserved in $x_p$ than the specialized ones, **WSAEs implicitly assign more weights to the general monosemantic features in the reconstruction loss**. This makes the general features more recoverable and accordingly prevents feature absorption.
>
> | Layer | 4 | 5 | 6 | avg |
> |--------------|--------------|-----------|-----------|-----------|
> | SAE  | 37.7 | 31.2 | 23.2 | 30.7 |
> | WSAE | **32.3** | **31.4** | **22.3** | **28.7** |
> | Gain (lower is better) | **-5.4** | +0.2 | **-0.9** | **-2.0** |
>
> ---
>
> **W3.** Figure 4 deliberately scales the x-axis to make minor differences in the semantic consistency metric look very large. It's unclear if this is meaningful or not.
>
> **A.** Thank you for pointing this out. We respectfully conjecture that the reviewer may be referring to the y-axis, where the semantic consistency metric is displayed. We would like to clarify that **the scaled x-axis in Figure 4 is only to improve visual readability**, and we will re-plot Figure 4 using a standard x-axis scale if necessary.
>
> In addition, we also note that **the differences in the semantic consistency metric are significant enough**. To address this, we conduct additional experiments with a larger range of $\alpha$ selection, and report the results in the following table. We show that the improvements of WSAEs over SAE are significant within a wide range of $\alpha$ choices.
>
> | $\alpha$ | 0 | 0.5 | 1.0 | 1.5 | 2.0 | 2.5 |
> |--------------|--------------|-----------|-----------|-----------|-----------|-----------|
> | semantic consistency  | 40.2 | 40.7 | 42.2 | 43.1 | 42.9 | 41.7 |
> | improvement over SAE  | 0 | **+0.5** | **+2.0** | **+2.9** | **+2.7** | **+1.5** |

---

> ### Author Response · Authors · 2025-11-21
>
> **Q1.** I'd like to see theoretical analysis which assumes more complex underlying data distributions. Would this change the resulting conclusions or insights?
>
> **A.** Following your suggestions, we have discussed the **hierarchical data distributions** of the latent features. (Please see our reply to **W1.** for more details.) This **does not change the insights of our theoretical conclusions**, and can also by part explain the feature absorption phenomenon empirically described in previous works.
>
> In addition, we also discuss a possible **non-linear generalization** of our underlying data distribution, which **also does not change the insights of our theoretical conclusions**. Specifically, we consider the attention-weighted combinations of features. We discuss that as $\mathrm{softmax}(\cdot)\approx \mathrm{max}(\cdot)$, the attention-weighted combinations can be approximately reduced to a linear form about certain features, so the theoretical results in our paper still hold. Please see the following for more details.
>
> For an attention-weighted combination of features, we denote the polysemantic feature as
> $$
> \boldsymbol{x}\_{p,t} = \sum\_{i=1}^n a_{t,i} v_i = \sum\_{i=1}^n \frac{(W_q\boldsymbol{x}_t)^\top (W_k \boldsymbol{x}_i)}{\sum_j (W_q\boldsymbol{x}_t)^\top (W_k \boldsymbol{x}_j)}W_v \boldsymbol{x}_i,
> $$
> where $W_q$, $W_k$, and $W_v$ denote query, key, and value matrices.
>
> Note that $\frac{(W_q \boldsymbol{x}\_t)^\top (W_k \boldsymbol{x}\_i)}{\sum\_j (W_q\boldsymbol{x}\_t)^{\top} (W_k \boldsymbol{x}\_j)}=\mathrm{softmax}((\boldsymbol{x}\_t^{\top} W_q^{\top} W_k \boldsymbol{x}\_j)\_{j\in [n]})\_i\approx \boldsymbol{1}[i=\arg\max_j \boldsymbol{x}\_t^\top W_q^\top W_k \boldsymbol{x}\_j]$, so we have
> $$
> \boldsymbol{x}\_{p,t} \approx \sum\_{i=1}^n \boldsymbol{1}[i=\arg\max_j \boldsymbol{x}\_t^\top W_q^{\top} W_k \boldsymbol{x}\_j] W_v \boldsymbol{x}\_i = W_v \boldsymbol{x}\_{\arg\max_j \boldsymbol{x}\_t^{\top} W_q^{\top} W_k \boldsymbol{x}\_j}.
> $$
> Then we have the SAE risk as
> $$
> \mathcal{L}\_{\mathrm{SAE}} = \mathbb{E}\_{\boldsymbol{x}} \sum\_{t=1}^n \\|\boldsymbol{x}\_{p,t} - W_m^{\top} \sigma(W_m\boldsymbol{x}\_{p,t})\\|^2 \approx \mathbb{E}\_{\boldsymbol{x}} \sum\_{t=1}^n \\|W_v \boldsymbol{x}\_{\arg\max_j \boldsymbol{x}\_t^{\top} W_q^{\top} W_k \boldsymbol{x}\_j} - W_m^{\top} \sigma(W_m W_v \boldsymbol{x}\_{\arg\max_j \boldsymbol{x}\_t^\top W_q^\top W_k \boldsymbol{x}\_j})\\|^2.
> $$
> By similar proofs of Theorems 1-3, if $W_v$ forms certain geometry or has non-positive interferences, then for $t \in [n]$, we have
> $$
> W_m^* = W_v^\top = \arg\min\_{W_m} \mathbb{E}\_{\boldsymbol{x}} \\|\boldsymbol{x}\_{p,t} - W_m^{\top} \sigma(W_m\boldsymbol{x}\_{p,t})\\|^2,
> $$
> and accordingly $W_m^* = W_v^\top = \arg\min_{W_m} \mathcal{L}_{\mathrm{SAE}}$.
>
> With this closed-form solution, we can show that the insights of SAE feature recovery still hold. Moreover, by similar proofs, the insights of Theorems 4-5 also still hold, only replacing $x$ with $\boldsymbol{x}_{\arg\max_j \boldsymbol{x}_t^\top W_q^\top W_k \boldsymbol{x}_j}$ in the mathematical forms.
>
> ---
>
> **Q2.** Use of established, if imperfect benchmarks like SAE bench could be used to provide a more comprehensive endorsement of the wSAE approach. For example, do wSAEs demonstrate less feature absorption?
>
> **A.** Following your suggestions, we conduct additional experiments on feature absorption to demonstrate the effectiveness of WSAE, and yes, WSAEs demonstrate less feature absorption. Please refer to our reply to **W2** for more details.
>
> ---
>
> Thank you for your insightful and constructive comments. We sincerely hope you can reconsider the recommendation score if our explanations and additional experiments solve your concerns. We also welcome further questions and discussions.

---

> ### Author Response · Authors · 2025-11-27
> **Could you please provide feedback on our rebuttal?**
>
> Dear Reviewer,
>
> We have done our best to address all your comments in the rebuttal. As the discussion period is ending in less than a week, we would greatly appreciate it if you could take a moment to review our responses and let us know whether our reply sufficiently resolved your concerns.
>
> Thank you very much for your time and consideration.
>
> Best regards,
>
> Authors

---

### Comment · Area_Chair_H1oQ · 2025-11-25
**Please discuss**

Several reviewers have not responded to the authors' rebuttals. Please read and respond to them.  Have the rebuttals addressed your concerns or clarified anything?

---

### Author Response · Authors · 2025-11-29
**Rebuttal Summary**

Dear Program Chairs, Senior Area Chairs, Area Chairs, and Reviewers,

We sincerely appreciate the tremendous efforts of the Program Chairs, Senior Area Chairs, and especially Area Chairs in coordinating the review process. We also extend our sincere thanks to all Reviewers for their constructive and detailed reviews.

We are greatly encouraged that the reviewers have positively recognized our paper’s **theoretical novelty, clarity, and significance** (Reviewers TYYe & hKP4), as well as **the principled motivation** (all reviewers) and **empirical effectiveness of WSAE** (Reviewer hKP4).

During the rebuttal phase, we have provided detailed point-by-point responses to all comments raised by the reviewers. We believe these discussions have significantly strengthened the paper. **Here, we provide a concise overview of the key improvements below:**

**1. Theoretical Clarifications and Extensions**

- **Clarifications on the modeling scope (Response to Reviewer DB94).** We clarify that the scope of our theoretical framework is to study whether a large enough SAE can recover the ground truth features, and our theoretical modeling is correct and appropriate under this scope.
- **Clarifications on the compatibility with dense latent features (Response to Reviewer DB94).** We clarify that our theoretical framework is compatible with dense latent features, the existence of which exactly motivates us to propose WSAE.
- **Extensions to hierarchical (overlapping) latent features (Response to Reviewer DB94).** We discuss extensions of our theoretical framework to hierarchical (overlapping) feature structures, which explains the feature absorption phenomenon that the reviewer is concerned about.
- **Extensions to multi-layer autoencoders (Response to Reviewer hKP4).** We discuss extensions of our theoretical framework to multi-layer linear autoencoders, which follow the same theoretical modeling as the hierarchical feature extensions. The theoretical conclusions are consistent with our original submission, that as long as the features are non-overlapping and extremely sparse, they can be fully recovered by SAEs.
- **Extensions to non-linear combinations of features (Response to Reviewers DB94 and hKP4).** We discuss extensions of our theoretical framework to attention-weighted combinations as an example of non-linear generalizations of our theoretical framework. Under this generalization, the theoretical conclusions are consistent with our original submission.
- **Extensions of theoretically inspired loss design beyond reweighting (Response to Reviewer TYYe).** We discuss alternative interference-reducing loss designs such as triangular-matrix estimation and regularization on $\\|\mathrm{trace}(W_p^{\top} W_p \Gamma^{\top}\Gamma)-n\\|$.
- **Theoretical link to dictionary learning and identifiability (Response to Reviewer TYYe).** We discuss that our theorems can be interpreted as a nonlinear extension of classical dictionary identifiability results, and that our proposed ground truth reconstruction loss can also be viewed as an identifiability measure.


**2. Additional Experiments**

- **SAEbench-style feature absorption (Response to Reviewer DB94).** We conduct additional experiments showing our WSAE consistently reduces feature absorption compared to SAE. We also provide a possible explanation that WSAE reduces feature absorption because it implicitly assigns more weights to the general monosemantic features.
- **$\alpha$-sweep robustness analysis (Response to Reviewers DB94 and TYYe).**
We conduct additional experiments for various $\alpha$ values, and show that WSAE maintains stable interpretability gains across all settings.
- **Proxy-robustness study (Response to Reviewers TYYe).** We conduct additional experiments to compare two monosemanticity proxies (semantic consistency vs. per-dimension variance) used in WSAE, and show that both yield consistent improvements over SAE.
- **Reconstruction-error Pareto check (Response to Reviewers TYYe).** We empirically verified that WSAE’s reconstruction error remains nearly identical to SAE across $\alpha$ values, confirming WSAE stays roughly on the Pareto frontier.
- **Large-model validation on Llama3-8B (Response to Reviewers hKP4).** We conduct additional experiments on the larger Llama3-8B model, showing WSAE improves interpretability across multiple layers, demonstrating scalability.

Beyond these major updates, we have also addressed other detailed inquiries in our individual responses, such as clarifying the extreme sparsity condition, clarifying the purpose of Figure 4 scaling, emphasizing our contributions within existing literature, and confirming reproducibility commitments.

**We respectfully hope that this summary is helpful.**

Thank you once again for your dedication to the community.

Best regards,

Authors

---

### Meta-Review · Area_Chair_7Aev · 2026-01-10

**Summary:**

The paper introduces the first theoretical framework for obtaining a closed-form solution for sparse autoencoders (SAEs). Using this framework, under assumptions on the data distribution they show that SAEs cannot recover ground truth monosemantic features unless they are extremely sparse. They also propose a remedy to boost reconstruction performance via a reweighting scheme.

This paper was received generally positively by reviewers, who praised the theoretical framework, the clarity and correctness of the technical exposition, and the potential impact of the new reweighting scheme on practical performance.

Some reviewers expressed concerns about whether the data distribution for which the theory holds was (un)realistic, questioned whether more complex distributions could be handled by the framework, asked clarification of novelty over existing theory for dictionary learning, and requested additional experimental analysis on phenomena such as feature absorption. Overall, the responses were (in my view) satisfactory.

Personally, I enjoyed reading the paper and congratulate the authors on the nice work. I would recommend incorporating reviewer feedback into future versions of the manuscript.

**Reviewer Concerns:**

The biggest criticism (and a totally fair one, in my view) was the (un)realistic assumptions on the data generating model. On the flip side, this is a criticism that can be leveled at most learning theory-flavored papers. The authors responded by outlining ways in which the theory could be extended to both more complex distributions as well as to deviations from the linear superposition model, which I thought was satisfactory.

The second criticism was on experimental evaluation. The authors responded by including new experiments on Pythia models following SAEBench, and showed promising results using their reweighting scheme. I thought this was satisfactory.

**Reviewer Scores:**

Reviewer DB94: this was the most negative review. However, I think the response was strong, and they may have improved their score after author discussion.

Reviewer TYYe: gave a good score but pointed out several issues. The responses were satisfactory, and they would have probably would have retained or improved their score.

Reviewer hKP4: gave a very strong score; would have retained it.

---

### Decision · Program_Chairs · 2026-01-26

Accept (Poster)